# Cohesin architecture and clustering in vivo

**Siheng Xiang, Douglas Koshland\***

Department of Molecular and Cell Biology, University of California, Berkeley, Berkeley, United States

**Abstract** Cohesin helps mediate sister chromatid cohesion, chromosome condensation, DNA repair, and transcription regulation. We exploited proximity-dependent labeling to define the in vivo interactions of cohesin domains with DNA or with other cohesin domains that lie within the same or in different cohesin complexes. Our results suggest that both cohesin's head and hinge domains are proximal to DNA, and cohesin structure is dynamic with differential folding of its coiled coil regions to generate butterfly confirmations. This method also reveals that cohesins form ordered clusters on and off DNA. The levels of cohesin clusters and their distribution on chromosomes are cell cycle-regulated. Cohesin clustering is likely necessary for cohesion maintenance because clustering and maintenance uniquely require the same subset of cohesin domains and the auxiliary cohesin factor Pds5p. These conclusions provide important new mechanistic and biological insights into the architecture of the cohesin complex, cohesin–cohesin interactions, and cohesin's tethering and loop-extruding activities.

**\*For correspondence:**
koshland@berkeley.edu

**Competing interests:** The authors declare that no competing interests exist.

## Introduction

Chromosome segregation, DNA damage repair, and the regulation of gene expression require the tethering or folding of chromosomes (*Uhlmann, 2016*; *Onn et al., 2008*). Remarkably, these different types of chromosome organizations are all mediated by a conserved family of protein complexes called structural maintenance of chromosomes (SMC) (*Onn et al., 2008*; *Nolivos and Sherratt, 2014*; *Hirano, 2016*; *Hassler et al., 2018*). SMC complexes tether and fold chromosomes by two activities. First, they can tether together two regions of DNA, either within a single chromosome or between sister chromatids (*Hassler et al., 2018*; *Onn et al., 2008*). Second, by combining this tethering activity with their ability to translocate along DNA, SMC complexes can also extrude DNA loops (loop extrusion) in vivo and in vitro (*Wang et al., 2017*; *Ganji et al., 2018*; *Davidson et al., 2019*; *Kim et al., 2019*). The molecular mechanisms for these activities and their regulation are still being elucidated.

SMC complexes have a conserved architecture (*Hassler et al., 2018*; *Figure 1A*). At their core, all SMC complexes have a dimer of two evolutionarily related proteins, which are called Smc (*Strunnikov et al., 1993*; *Hirano and Mitchison, 1994*; *Strunnikov et al., 1995*). Each Smc subunit has two globular domains, a head domain and a hinge domain, separated by a long 50-nm coiled coil. Smc subunits dimerize by two distinct interactions, one between the two heads and another between the two hinges. The separation of head and hinge dimers by their intervening long coiled coils allows the Smc dimers to achieve multiple conformations in vitro, including rings, rods, or butterflies (*Figure 1A*). The butterfly conformations result from bending of the coiled coils of the SMC subunits bringing the hinge and head domains in close proximity (*Hirano et al., 2001*; *Soh et al., 2015*; *Bürmann et al., 2019*). The Smc subunits also act as scaffolds to bind non-Smc proteins (*Yatskevich et al., 2019*). The presence of different complex architectures in vitro raises many questions. Which, if any, of these different conformations exist in vivo? How do any of these conformations contribute to different SMC functions? Are there other structural features of SMC complexes,

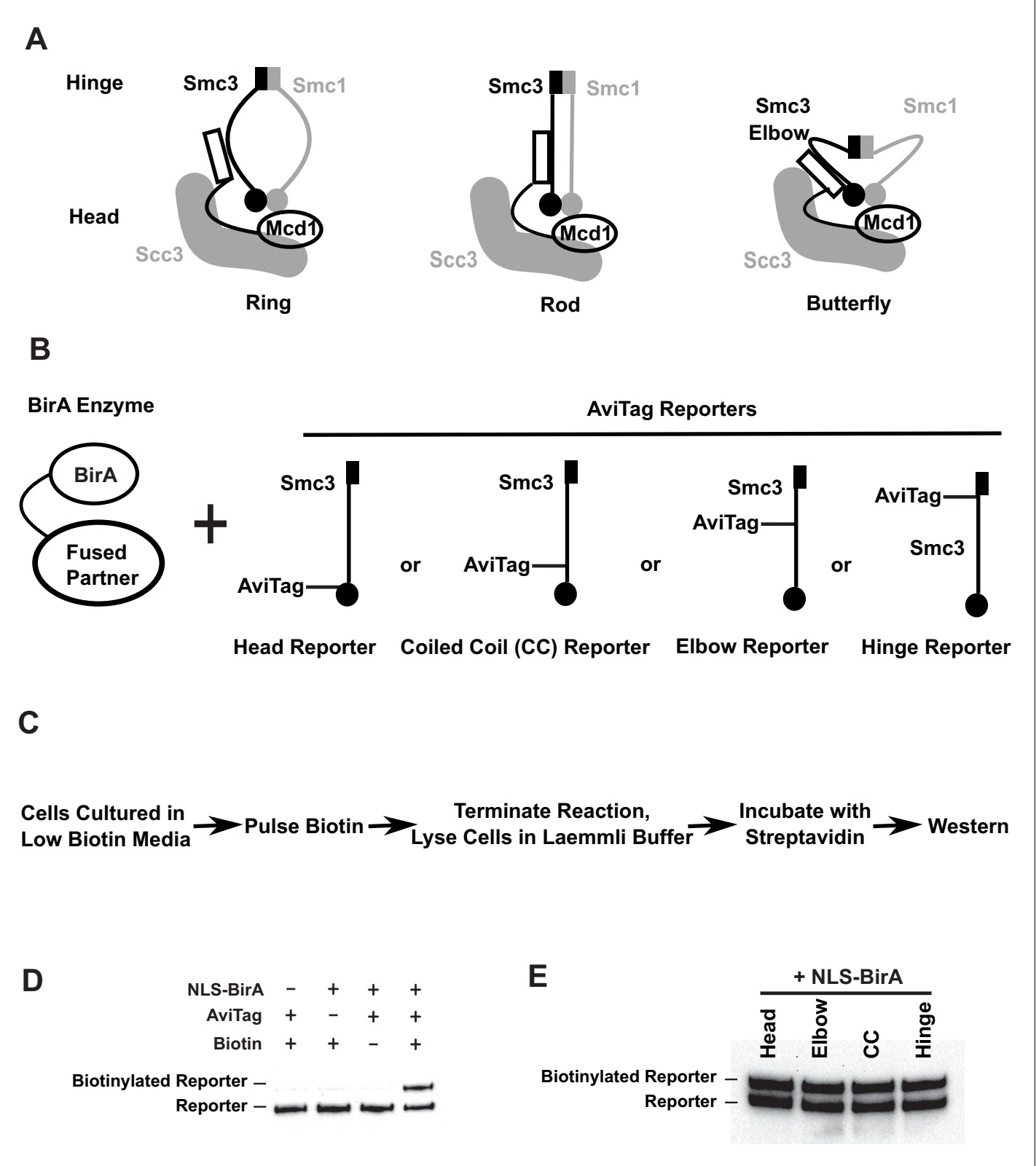

**Figure 1.** Smc3p reporters for proximity biotinylation experiments. (**A**) Cartoons showing the proposed ring, rod, or butterfly conformation of budding yeast cohesin. Each cohesin complex consists of one single copy of each of the four subunits: Smc1p, Smc3p, Mcd1p, and Scc3p. (**B**) Cartoons depicting strains used in proximity biotinylation experiments. Each strain expresses one of the four AviTag reporters and a BirA enzyme fused to a partner. Each reporter had an AviTag fused to a flexible linker of 6xHA tag inserted into the head (head reporter), the joint of coiled coil (CC reporter),

*Figure 1 continued on next page*

*Figure 1 continued*

the elbow-proximal coiled coil (elbow reporter), or the hinge (hinge reporter). (**C**) Experimental regime of proximity biotinylation assays. Cells are cultured in synthetic medium with low biotin and pulsed with biotin briefly. The biotinylation reactions are terminated with the addition of trichloroacetic acid (TCA), and cell lysates are prepared in the Laemmli buffer. The samples are then diluted and incubated with streptavidin and subjected to western blot analysis using anti-HA antibodies to detect both biotinylated and non-biotinylated AviTag reporter. (**D**) Streptavidin gel shift assays (see 'Streptavidin gel shifts' in 'Materials and methods') to detect biotinylation of the head reporter. Cells were cultured in low biotin synthetic medium with galactose and arrested in mid-M phase by nocodazole. Cell lysates were prepared in sodium dodecyl sulfate (SDS) sample buffer, incubated with 1 mg/ml streptavidin for 15 min and biotinylation of the head reporter is seen as a slower migrating band (lane 4, SX80F). No reporter biotinylation was detected in cells missing BirA enzyme (lane 1, SX73), in cells with a dead AviTag (lane 2, SX222, no lysine in AviTag), or in cells treated with TCA before biotinylation (lane 3, SX80F). (**E**) Comparison of AviTag accessibility to BirA-mediated biotinylation. Cultures of strains with *SMC3* reporter alleles (SX80B, SX80D, SX80E, and SX80F) were grown in low biotin synthetic media at 30°C overnight and treated with nocodazole for 2.5 hr to arrest cells in mid-M phase. Then NLS-BirA, a nuclear-localized version of BirA enzyme, was induced for 1 hr. About 10 nM biotin was added to initiate biotinylation. After a 7-min biotin pulse, the reactions were terminated by TCA. Efficiencies of reporter biotinylation were assessed by streptavidin gel shift.

The online version of this article includes the following figure supplement(s) for figure 1:

**Figure supplement 1.** Tagged SMC3 alleles support cell viability.

like clustering (dimers or multimers)? If clusters exist, what is their function, and what factors control their formation?

One SMC complex, called cohesin, was originally discovered in budding yeast because it mediates cohesion between sister chromatids (*Guacci et al., 1997*; *Michaelis et al., 1997*). The subunits of budding yeast cohesin (*Figure 1A*; Smc1p and Smc3p, and the non-SMC subunits Mcd1p/Scc1p and Scc3p) are conserved throughout eukaryotes (*Guacci et al., 1997*; *Michaelis et al., 1997*; *Yatskevich et al., 2019*). Sister chromatid cohesion is essential for chromosome segregation in mitosis and meiosis (*Guacci et al., 1997*; *Michaelis et al., 1997*; *Klein et al., 1999*). Subsequently, cohesin was also shown to be important for chromosome condensation, efficient DNA repair, and the regulation of gene expression (*Guacci et al., 1997*; *Ström et al., 2004*; *Wood et al., 2010*; *Dorsett and Ström, 2012*; *Lopez-Serra et al., 2013*; *Bloom et al., 2018*; *Zhang et al., 2019*; *Lamothe et al., 2020*). Mutations in cohesin subunits are thought to drive cancer and cause age-dependent congenital disabilities and developmental disorders (*Brooker and Berkowitz, 2014*; *Romero-Pérez et al., 2019*; *Ogawa, 2019*; *Watrin et al., 2016*; *Chiang et al., 2010*).

To promote proper chromosome segregation, cells must establish cohesion in S phase. Cohesion establishment requires two steps. First, the Scc2p/Scc4p complex loads cohesin on one sister chromatid (*Ciosk et al., 2000*). Then cohesin is acetylated by Eco1p, which promotes tethering of the second sister chromatid (*Tóth et al., 1999*; *Skibbens et al., 1999*; *Unal et al., 2008*; *Rolef Ben-Shahar et al., 2008*; *Guacci and Koshland, 2012*; *Çamdere et al., 2015*).

Proper chromosome segregation also requires that cohesion be maintained from S phase to M phase (*Hartman et al., 2000*). Cohesin acetylation helps to maintain cohesion by preventing cohesin from being removed by Wpl1p from chromosomes (*Kueng et al., 2006*; *Rolef Ben-Shahar et al., 2008*; *Lopez-Serra et al., 2013*). Cohesion maintenance is also promoted by the cohesin binding protein called Pds5p (*Hartman et al., 2000*; *Tanaka et al., 2001*; *Wang et al., 2002*; *Noble et al., 2006*). Pds5p facilitates efficient Eco1p-dependent acetylation of cohesin, thereby inhibits Wpl1p-dependent cohesin release (*Chan et al., 2013*). In addition, Pds5p maintains cohesion by a second unknown mechanism that is independent of Wpl1p inhibition (*Tanaka et al., 2001*).

The complex activities and regulation of cohesin likely depend on the presence and modulation of one or more of its conformations. The ring, rod, and butterfly conformations of cohesin (*Figure 1A*) have been interrogated both in vitro and in vivo. Cohesin rings have been observed by electron microscopy (EM) and liquid atomic force microscopy (AFM) (*Hirano et al., 2001*; *Anderson et al., 2002*; *Ryu et al., 2019*). Rings have also been inferred to exist in vivo from experiments showing cohesin's ability to topologically entrap DNA (*Haering et al., 2008*). Rods have been seen by EM and inferred to exist in vivo from cross-linking experiments (*Anderson et al., 2002*; *Soh et al., 2015*); however, rods were not observed in liquid AFM images (*Ryu et al., 2019*). A butterfly conformation forms in vitro by bending of the cohesin coiled coil at the elbow (*Bürmann et al., 2019*). While the butterfly structure is extremely rare in negatively stained electron microscopies, it was more prevalent in liquid AFM images (*Ryu et al., 2019*). The existence of a

butterfly structure in vitro is also supported by the detection of interactions between purified Scc3p and the hinge in recent cryo-EM images of cohesin (*Murayama and Uhlmann, 2015*; *Shi et al., 2020*; *Higashi et al., 2020*). In vivo, the existence of this structure has been inferred from imaging and functional interactions of the hinge and the head-bound Pds5p (*Mc Intyre et al., 2007*; *Eng et al., 2015*). Therefore, evidence for multiple cohesin conformations exists both in vitro and in vivo, but their physiological significance and relative abundances in vivo are unclear.

Like cohesin conformation, cohesin–cohesin interactions have also been investigated in vitro and in vivo. In two recent single molecular studies, the photobleaching of fluorescently tagged cohesin came to different conclusions about whether cohesin bound to DNA as monomers or dimers (*Kim et al., 2019*; *Davidson et al., 2019*). In vivo, the presence of dimers in cells was tested by co-immunoprecipitation. Other studies also disagree about the presence of dimers (*Zhang et al., 2008*; *Haering et al., 2002*). More recent in vivo studies revealed the functional complementation between distinct cohesin complexes that are individually defective on their own (*Eng et al., 2015*; *Srinivasan et al., 2018*). This complementation could be best explained by the clustering of cohesins. These discrepancies make physiological significance and the relative abundance of cohesin clusters unclear. To begin to answer these many unanswered questions about cohesin clustering and architecture, we interrogated cohesin with the method of proximity-dependent biotinylation (*Fernández-Suárez et al., 2008*; *Jan et al., 2014*).

## Results

### Developing a proximity biotinylation method to assess cohesin structure and clustering in vivo

Proximity biotinylation is a method that exploits the ability of the bacterial enzyme, BirA, to recognize a 15 amino acid AviTag and add a biotin moiety to a lysine residue within the AviTag (*Fernández-Suárez et al., 2008*; *Branon et al., 2018*). In vivo, biotinylation of the AviTag is enhanced when the AviTag and BirA are fused to two interacting partners that place the BirA and AviTag in proximity. Importantly, the BirA and AviTag partners must be in very close proximity to generate AviTag biotinylation because this biotinylation only occurs when the AviTag binds to the active site of BirA.

Given that cohesin is a large complex, the levels of biotinylation of the AviTags inserted into different domains of cohesin should reflect their relative proximity to a BirA partner. We inserted AviTags (marked with an HA epitope) at each of the four positions in Smc3p of *Saccharomyces cerevisiae*. These AviTag reporters were positioned in the head domain (head reporter, in the flexible loop after residue A1089), within the coiled coil proximal to the head (CC reporter, in the joint after residue V966), at the elbow-proximal coiled coil (elbow reporter, after residue F864), and in the hinge domain (hinge reporter, after residue P533) (*Figure 1B*, see *Figure 1—figure supplement 1B* for tag insertion sites in the elbow folded structure suggested in *Bürmann et al., 2019*). We also generated three BirA partners (*Figure 1B*) by inserting the BirA enzyme (marked with V5) in the Smc3p head domain (after Smc3p residue A1089), in the Smc3p hinge domain (after Smc3p residue P533), and at the C-terminus of the HUα DNA binding protein from bacteria. The biotinylation of an AviTag by a BirA partner inserted in the same molecule is defined as *cis*-biotinylation. The biotinylation of an AviTag by a BirA partner inserted in another molecule is defined as *trans*-biotinylation.

Using these reagents, we developed an assay for BirA-based proximity-dependent biotinylation of Smc3p reporters in vivo (*Figure 1C*). Strains were constructed harboring different AviTag and BirA partners. Cultures of these strains were grown in media lacking biotin and then subjected to a short pulse of biotin to initiate biotinylation. The biotinylation was terminated by adding TCA. The purpose of this short pulse of biotin was dual. It allowed us to limit biotinylation to specific times in the cell cycle to test temporal regulation of the interactions between our reporters and a BirA-tagged partner. The biotin pulse also suppressed proximity-independent biotinylation of the reporters that occurs with extensive incubation with biotin. After the biotin pulse, protein extracts were prepared, and the biotinylated and non-biotinylated Smc3p-AviTag was detected by western blotting using the HA epitope. The biotinylated Smc3p-AviTag was present as a slower mobility species due to its binding to the streptavidin present in the protein sample buffer (*Figure 1C, D*).

We first validated the gel shift as a measure of biotinylation of the AviTag by BirA in vivo. We examined the requirements for Smc3p-AviTag gel shift with one of our Smc3-AviTag reporters and a

freely diffusible nucleoplasmic BirA. No biotinylation of the Smc3p-AviTag reporter was observed before the biotin pulse, in cells lacking BirA, or in cells expressing a reporter in which the critical Avi-Tag lysine was mutated (*Figure 1D*). We observed a gel shift of the Smc3p-AviTag reporter only upon the simultaneous presence of the AviTag, BirA, and biotin (*Figure 1D*). We conclude that the observed gel shift indeed reflected the biotinylation of the AviTag by BirA in vivo.

We performed three additional control experiments to ensure that we could properly interpret differences in biotinylation of the AviTag reporters. First, we showed that all the Smc3p-AviTag reporters and Smc3p-BirA were functional in vivo. Strains bearing any of these *SMC3-AviTag or SMC3-BirA* genes as the sole source grew at similar rates as the wild-type strain (*Figure 1—figure supplement 1*). Furthermore, we showed that all four Smc3p-AviTag reporters are biotinylated to the same extent by freely diffusible nucleoplasmic BirA (*Figure 1E*). This result established that any differences in their levels of biotinylation by a BirA partner reflected differences in the association with that partner rather than the inherent limitations of the reporters. We also showed that our western blot-based assay was quantitative because the Western signal of the Smc3p-AviTag reporter showed a linear response to serial dilutions of Smc3-AviTag protein (*Figure 1—figure supplement 1B*). Thus, any differences between the levels of reporter biotinylation by a BirA partner reflected relative differences in the proximity of the reporters to that partner.

## The structures of Smc heterodimer and cohesin tetramer are highly dynamic in vivo

To assess the intramolecular interactions between domains of Smc3p, we constructed five strains. Each strain carried a single *SMC3* gene that encoded a doubly tagged Smc3p with one of its domains marked by an AviTag and another by a BirA partner (an example is presented in *Figure 2A*). We reasoned that intramolecular biotinylation (*cis*-biotinylation) of an AviTag by its BirA partner reflected the bringing together of the AviTag- and BirA-marked domains within an Smc3p subunit.

We began by assessing the intramolecular interactions of Smc3p domains within Smc3p-Smc1p heterodimers that are present in G1 cells. G1 cells in budding yeast assemble the Smc3p-Smc1p heterodimers rather than the full cohesin complex because Mcd1p is not expressed, and Scc3p cannot bind to the heterodimer without Mcd1p (*Haering et al., 2002*). The heterodimers lie in the nucleoplasm because they are unable to bind DNA (*Tanaka et al., 1999*). We observed that BirA in the Smc3p hinge domain robustly biotinylated (65–90%) AviTags in the elbow-proximal coiled coil, the head-proximal coiled coil, and the head (*Figure 2C, Figure 2—figure supplement 1C*). The biotinylation of the AviTags likely occurred by interactions with the BirA partner in cis because of their close proximity.

However, some AviTag biotinylation in these strains could result from *trans*-biotinylation (binding and biotinylation of the AviTag on one Smc3p by BirA on another Smc3p) as well as *cis*-biotinylation. To assess the potential contribution of *trans*-biotinylation in G1, we constructed a strain with two copies of *SMC3*, one with the BirA in the head domain and the other with the AviTag in the hinge. The biotinylation of hinge Smc3p in G1 was reduced 87% compared to when BirA and AviTag were in the head and hinge of the same Smc3p (*Figure 2B, Figure 2—figure supplement 1B*). We conclude that the majority of the biotinylation that we observe for Smc3p in the Smc3p-Smc1p heterodimers reflects the interaction of the partner domains in cis. We present additional data supporting the biotinytlation of doubletagged Smc3p occurs in cis for the fully assembled cohesin complex (see section entitled "Pds5p is required for cohesin clustering on and off chromosomes").

The promiscuous *cis*-biotinylation pattern of the three AviTags by the hinge BirA strongly suggests that the heterodimer is structurally dynamic, transitioning between multiple conformations in vivo. If the heterodimer folded into a single rigid conformation (ring, rod, or butterfly), only one or no AviTag site would be close enough to bind to the BirA active site and be biotinylated. We suggest that the coiled coil between the hinge and elbow is flexible, allowing the hinge domain to bind to the elbow-proximal coiled coil, coiled coil, or head domains, reflecting multiple distinct butterfly conformations.

To test the robustness of this conclusion, we examined the interactions of the AviTag reporters with a BirA partner in the head. The head BirA robustly biotinylated head-proximal coiled coil and the hinge AviTags. The ability of the head BirA to biotinylate these two AviTags also is consistent with a folded but structurally dynamic heterodimer.

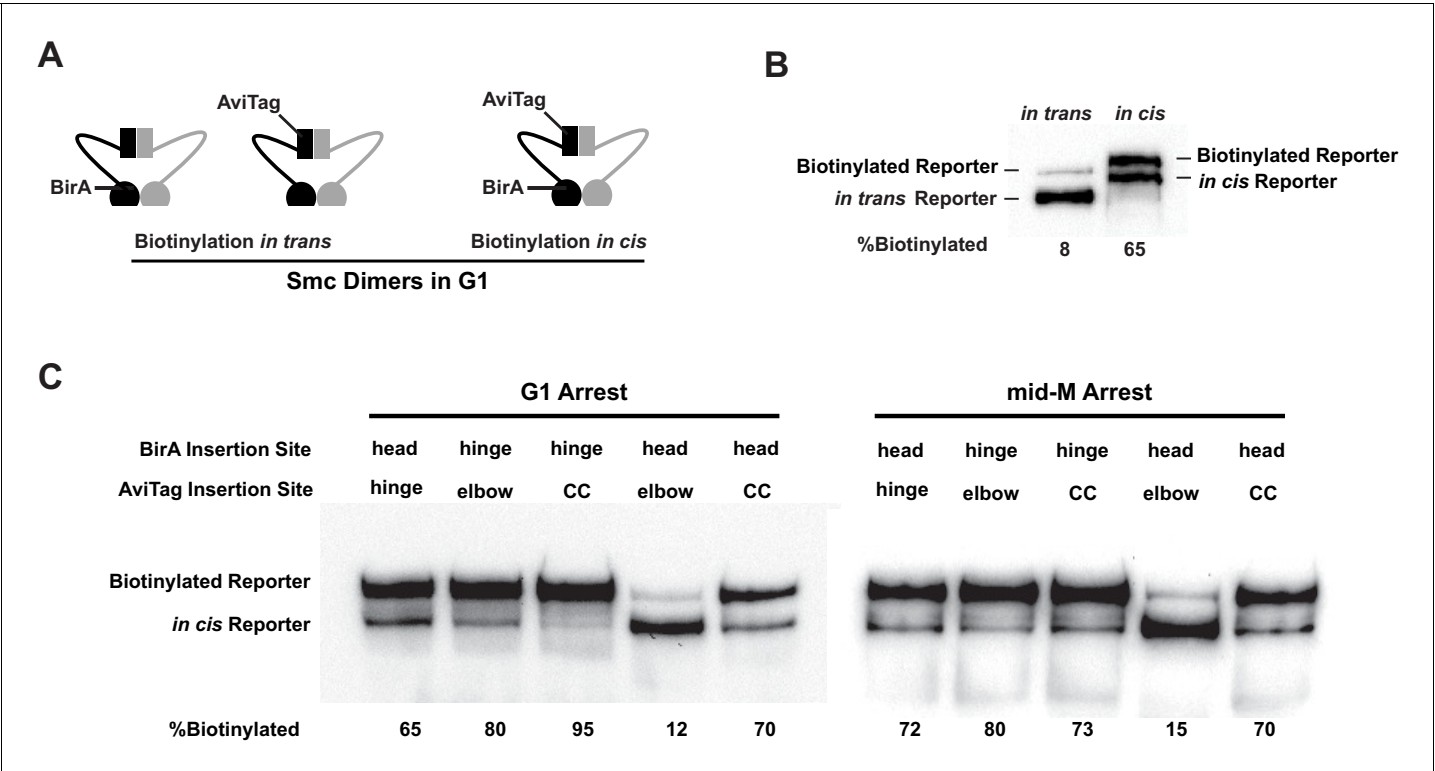

**Figure 2.** Cohesin adopts a dynamic butterfly conformation in vivo. (**A**) Cartoons depicting inter- and intramolecular biotinylation of Smc3 hinge reporter. Both strains were cultured to early log phase and arrested in G1, where cohesin complexes are not fully assembled. In trans, biotinylation experiment was carried out in a strain carrying two *SMC3* alleles; one allele was tagged with BirA enzyme in the head domain, while the second allele was tagged with AviTag in its hinge. In cis, biotinylation was carried out in the cells expressing double-tagged Smc3, with BirA in the head and AviTag inserted in the hinge domain of the same molecule. Mcd1p and Scc3p are omitted from the cartoon for clarity. (**B**) Comparison of biotinylation levels of Smc3 reporters in cis (SX246) and in trans (SX48B). Streptavidin gel shift assay as described in *Figure 1D* was used to compare Smc3p-AviTag biotinylation in trans and in cis. (**C**) Smc dimers in G1 and cohesin tetramers in mid-M arrested cells adopt the butterfly conformation. Each strain carries a *cis*-biotinylation reporter allele of SMC3 (SX246, SX247, SX248, SX249, and SX250). The strain was arrested in G1 with α-factor or mid-M using nocodazole. The cells were then treated with biotin pulse, and biotinylation efficiencies were assayed by streptavidin gel shift.

The online version of this article includes the following figure supplement(s) for figure 2:

**Figure supplement 1.** Cartoon of elbow folded Smc3p and quantitative analysis of *Figure 2*.

However, the head BirA, unlike the hinge BirA, poorly biotinylated the elbow reporter. Its biotinylation (12%) was 80–90% lower compared to the other AviTag reporters. This suppression of biotinylation could not be explained by the elbow reporter being generally inaccessible to BirA. The elbow reporter could be *cis*-biotinylated by the hinge BirA or *trans*-biotinylated by freely diffusible BirA to the same high level as the other AviTag reporters (*Figure 1E, Figure 2C, Figure 2—figure supplement 1C*). We conclude that the head BirA, unlike the hinge BirA, must be constrained to prevent its interaction with the elbow reporter. We suggest that the coiled coil between the head and elbow is stiffer than the coiled coil between the hinge and elbow, thereby limiting the ability of the head domain to interact with the elbow.

We wondered whether the intramolecular interactions of Smc3p domains were altered by the assembly of the full cohesin complex or cohesin binding to DNA. Therefore, we examined the biotinylation pattern for Smc3p reporters in this same set of strains in mid-M (nocodazole arrested) where the cohesin was assembled and bound to DNA (*Figure 2C, Figure 2—figure supplement 1C*). The biotinylation patterns were indistinguishable for the Smc3p in the fully assembled cohesin complex in mid-M and the Smc3p-Smc1p heterodimers in G1. These results are consistent with the conclusion that the fully assembled cohesin complex, like the heterodimers, is capable of forming multiple butterfly conformations in vivo.

### The hinge and head domains are proximal to DNA in vivo

To test which domain(s) of Smc3p are proximal to DNA, we constructed a set of strains expressing one of our Smc3p-AviTag reporters along with the BirA enzyme fused to HUα, a non-specific DNA binding protein from bacteria (*Rouviere-Yaniv and Gros, 1975*; *Figure 3A*). The different strains were cultured in low biotin media, arrested in G1, then released into media containing nocodazole. These cells synchronously progressed through S phase and then were arrested in mid-M phase where a large fraction of cohesin is stably bound to chromosomes to promote cohesion. The mid-M arrested cells were treated with a biotin pulse and then assessed for reporter biotinylation (*Figure 3B*). The BirA fused to the DNA-binding protein HUα biotinylated the hinge and head reporters to similar levels (about 18–20%), while only 10% and 4% of the elbow and head-proximal coiled coil reporters, respectively, were biotinylated (*Figure 3C*, left, *Figure 3—figure supplement 1B*). These results suggest that the head and the hinge domains are more proximal to chromosomal DNA than the elbow and the CC.

To test whether this biotinylation pattern was due to the interaction of the DNA-bound cohesin with DNA-bound HUα-BirA, we repeated the experiments in cells where cohesin binding to DNA was prevented. Scc2p, a subunit of cohesin loader, is essential for loading cohesin onto DNA. With this in mind, we introduced an auxin degron allele of *SCC2* (*SCC2-AID*) into our set of strains. Cultures of these *SCC2-AID* strains were arrested in G1 phase, where the cohesin loader was depleted

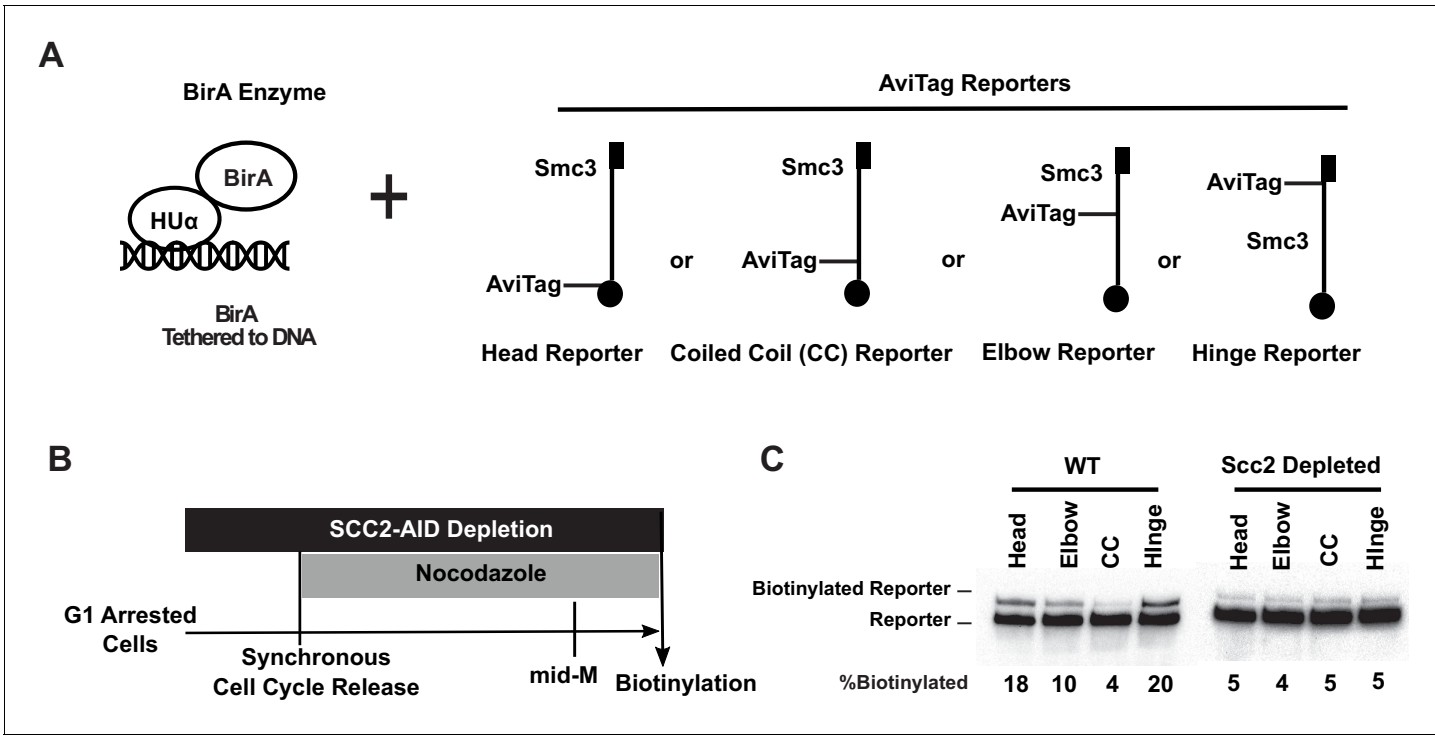

**Figure 3.** Cohesin binds DNA with both head and hinge domains on DNA. (**A**) Cartoons depicting strains used in this figure. BirA enzyme was tethered to DNA by HUα, a bacterial non-specific DNA binding protein. This DNA-tethered BirA was expressed in cells carrying a head, elbow, coiled coil (CC), or hinge reporter Smc3, respectively. Mcd1p and Scc3p are omitted from the cartoon for clarity. (**B**) Experimental regimen used to test cohesin architecture on DNA. Cells were cultured in synthetic media with low biotin overnight to early log phase and arrested in G1 by α-factor. About 1 mM auxin was added to the arrested cells to deplete the cohesin loader in *SCC2-AID* cells. Cells were synchronously released from G1 and arrested in mid-M phase in low biotin media containing auxin and nocodazole, then the cultures were treated with biotin pulse. Half of the cells were analyzed by western blotting to assess biotinylation efficiencies of the Smc3 reporters. The other half of the cultures were fixed to assay cohesin on DNA by chromatin immunoprecipitation (*Figure 3—figure supplement 1A*). (**C**) Biotinylation of Smc3 reporters by DNA-tethered BirA. The gel on the left side shows biotinylation of Smc3 reporters in wild-type cells (SX136B, SX136D, SX136E, and SX136F). The gel on the right side shows biotinylation of Smc3 reporters in cells depleted of loader subunit Scc2p (SX221B, SX221D, SX221E, and SX221F).

The online version of this article includes the following figure supplement(s) for figure 3:

**Figure supplement 1.** Confirmation of Scc2p-AID depletion and quantitative analysis of *Figure 3*.

by adding auxin. The cultures synchronously released into nocodazole and auxin, allowing them to progress through S phase and then arrest in mid-M phase without Scc2p (*Figure 3B*). Half of the arrested cultures were fixed with formaldehyde, and cohesin association with DNA was assessed by chromatin immunoprecipitation (ChIP) using an antibody specific for the endogenous cohesin subunit Mcd1p. The other half of the culture was subject to a short biotin pulse to assess HUα-BirA-mediated biotinylation of the reporters.

As expected, the Scc2p depletion eliminated the Mcd1p ChIP signal at a representative cohesin-associated region (CAR) (*Figure 3—figure supplement 1A*), confirming that cohesin loading onto chromosomes had been abolished. Importantly, we found that all three Smc3p reporters exhibited only basal levels of biotinylation (4–5%) under these conditions (*Figure 3C*, right, *Figure 3—figure supplement 1B*). Cohesin binding to DNA enhanced head and hinge reporter biotinylation four-fold. The biotinylation of elbow reporter was enhanced only two-fold, whereas head-proximal coiled coil reporter biotinylation was not enhanced. This hierarchical biotinylation of cohesin domains by DNA-bound BirA suggests that the hinge and head domains are proximal to the DNA, while the elbow-proximal coiled coil and the head-proximal coiled coil are more distal.

## Cohesin forms ordered clusters in vivo whose levels are regulated during the cell cycle

Cohesin may form disordered clusters or ordered clusters (like oligomers) in vivo. If clustering occurs by either mechanism, two domains from different cohesin complexes should be proximal to each other. To study cohesin clustering using proximity biotinylation, we constructed three strains, each expressing two differentially tagged Smc3p. All strains have one copy of Smc3p with BirA inserted into its head domain. The second copy of Smc3p in each strain has the AviTag reporter inserted at a different location, either in the head, in the CC, or in the hinge domains (*Figure 4A*). To probe for clustering during the cell cycle, cells were cultured in low biotin media, arrested in G1, S, or mid-M phase, and then subjected to a biotin pulse. Cell cycle arrests were confirmed by flow cytometry analysis of the DNA content (*Figure 4—figure supplement 1A*). BirA can only biotinylate an AviTag when the enzyme touches its substrate directly. Therefore, preferential biotinylation of an Smc3p reporter would occur in cohesin clusters when the AviTag-tagged domain was close to the head domain BirA in the second Smc3p (Smc3p-BirA). The maximum level of biotinylation of the Smc3p reporter would be 50% for an ordered cluster (a dimer or a multimer, like microtubule) where the domain of one cohesin touches the domain(s) of only one neighboring cohesin.

Biotinylation levels of all three Smc3p reporters were low in G1 (α-factor) arrested cells. *Trans*-biotinylation of the head and hinge reporters increased 7.5-fold in S phase (hydroxyurea) arrested cells to 45% and four-fold in mid-M phase (nocodazole) arrested cells to 25% (*Figure 4C, Figure 4—figure supplement 1B*). As expected for *trans*-biotinylation, no biotinylation of the Smc3p reporter was observed before the biotin pulse, in cells lacking BirA, or in cells expressing a reporter in which the critical AviTag lysine was mutated (*Figure 4—figure supplement 1F*). These results show that cohesin efficiently clusters in vivo.

The *trans*-biotinylation of the CC reporter by the BirA in the head of another cohesin molecule remained low in S and mid-M phase arrested cells. This low level was not due to inherent inaccessibility of the CC to any BirA partner as high levels of CC reporter biotinylation were seen using free nucleoplasmic BirA in trans or by the head or hinge BirA in cis. These results are consistent with cohesin clustering in an ordered manner that enables head–head and head–hinge interactions between different cohesin molecules, but disfavors interaction between CC of one cohesin complex and the head of another cohesin.

To test whether these cell cycle differences also occurred in normally dividing cells, we assayed biotinylation in cells synchronously dividing after release from G1 arrest. Cells carrying both the *SMC3-BirA* and the *SMC3-AviTag* hinge reporter alleles were cultured in low biotin media, arrested in G1, and synchronously released into a mid-M arrest. Aliquots of the culture were taken every 20 min, and half of the cells were treated with biotin pulse to assay Smc3p-AviTag reporter biotinylation (*Figure 4—figure supplement 1C*). The other half of the cells from each aliquot was fixed, and DNA content in the cells was analyzed by flow cytometry. Cells exhibited a basal level of biotinylation 20 min after release from G1 (*Figure 4—figure supplement 1D*). By 40 min some cells started entering S phase (*Figure 4—figure supplement 1E*), and biotinylation levels increased and peaked at 60–80 min, the latter time marked the end of S phase. Subsequently, biotinylation decreased about 50% to

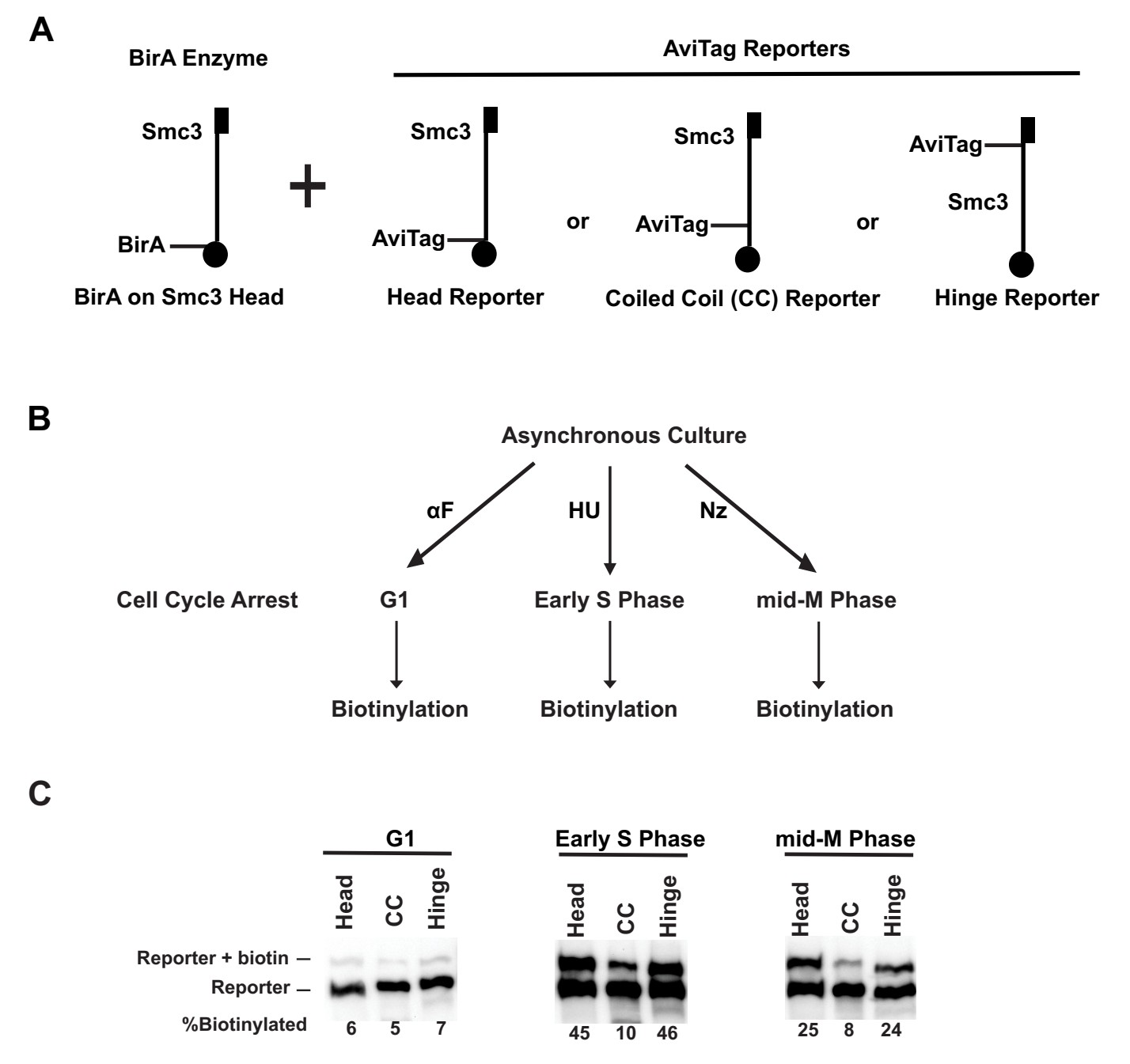

**Figure 4.** Cohesin forms clusters that are cell cycle-regulated. (**A**) Cartoons depicting the strains used (SX48B, SX48E, and SX48F). Each strain carries two *SMC3* alleles. One *SMC3* allele is tagged with BirA enzyme in its head domain. The other *SMC3* allele is the reporter, with AviTag fused to the head domain, the coiled coil (CC), or the hinge domain. Mcd1p and Scc3p are omitted from the cartoon for clarity. (**B**) Experimental regimen to assess cohesin clusters as a function of the cell cycle. Asynchronous cultures were treated with α-factor, hydroxyurea, or nocodazole to arrest cells in G1, early S phase, or mid-M, respectively. Cells were then treated with a biotin pulse to access levels of cohesin clusters. (**C**) Comparison of cluster formation in G1, S, and mid-M. Asynchronous cultures expressing Smc3p-BirA and one of the three Smc3p reporters as shown in *Figure 3A* were treated as described in *Figure 3B* to assess intermolecular biotinylation of the reporters. Biotinylation was assessed by streptavidin gel shift assay as described in *Figure 1D*.

The online version of this article includes the following figure supplement(s) for figure 4:

**Figure supplement 1.** Cohesin clusters in cells synchronously released from G1 into mid-M phase.

a similar biotinylation level as observed in cells arrested in mid-M (*Figure 4—figure supplement 1D*, *Figure 4C*). The results from arrested and cycling cells demonstrate that cohesin clustering is cell cycle-regulated, forming and peaking during S phase and then partially dissolving from G2 to M phase. The presence of clusters between S phase to mid-M correlates with the time when cohesion is established and maintained.

## Cohesin clusters are bound to DNA and their levels at CARs are cell cycle-regulated

Next, we asked whether cohesin clusters are present on chromosomes. Cohesin binds at centromeres, pericentric regions, and distinct peaks every 10–15 kb along chromosome arms called CARs (*Blat and Kleckner, 1999*; *Laloraya et al., 2000*; *Lengronne et al., 2004*; *Glynn et al., 2004*). We asked whether in *trans*-biotinylation of the Smc3p-AviTag reporters by the Smc3p-BirA could be detected on chromosomes.

To detect the presence and localization of *trans*-biotinylated Smc3p-AviTag on DNA, we modified our AviTags and the readout after biotinylation in vivo. We co-expressed Smc3p-BirA with a modified Smc3p head reporter in which the AviTag was placed at the end of a Myc-tag linker. This intervening linker made the biotinylated AviTag more accessible to immunoprecipitation by streptavidin beads (*Figure 5A*). These tags did not affect the chromatin localization of cohesin. As shown in *Figure 5—figure supplement 2*, both BirA-tagged and AviTag-tagged Smc3p, detected using anti-V5 and anti-MYC antibodies, respectively, robustly localized at previously identified CARs at the centromere, at a pericentric site, and on chromosomal arms to similar levels. After pulsing these cells with biotin, we used streptavidin beads to enrich DNA sequences that were bound to Smc3p-AviTag that had been biotinylated in trans by Smc3p-BirA ('Materials and methods'). DNA sequences associated with the biotinylated Smc3p marked the position of cohesin clusters on chromosomes.

We began by assessing the levels and positions of DNA-bound clusters in S phase. The cells were arrested in S phase (hydroxyurea), pulsed with biotin, then fixed and processed for ChIP using streptavidin beads (*Figure 5B*). We detected biotinylated Smc3p reporter at centromeres (e.g., *CEN3*), at pericentric regions, and at arm CARs using qPCR (*Figure 5C*, *Figure 5—figure supplement 1A*). These ChIP signals were specific to reporter biotinylation as they were absent or dramatically reduced in cells lacking BirA, that is, expressing only Smc3p-AviTag without Smc3p-BirA. We repeated these experiments using a strain bearing the same Smc3p-AviTag reporter but overexpressing the free nuclear BirA (NLS-BirA) that we showed maximally biotinylated all the Smc3p-AviTag reporters (*Figure 1E*). The ChIP signals of biotinylated Smc3p head reporter by the NLS-BirA and that by Smc3p-BirA were similar at all CARs tested (*Figure 5C*, *Figure 5—figure supplement 1A*). This similarity indicates that cohesin molecules at all CARs tested are clustering in S phase.

We wondered whether the distribution of chromosome-bound cohesin clusters might change through the cell cycle, given the cell cycle-regulated changes in cluster abundance. We assayed for the presence of cohesin clusters in the mid-M phase (nocodazole) arrested cells (*Figure 5D*, *Figure 5—figure supplement 1B*). As in S phase arrested cells, Smc3p-BirA-mediated *trans*-biotinylation of the Smc3p-AviTag reporter was robust at both *CEN* and pericentric regions. However, the *trans*-biotinylation signal was greatly reduced at some arm CARs but still above the signal in the negative control of cells lacking cells without BirA. At other CARs, this signal was eliminated. The loss of ChIP signals was not due to loss of binding between the Smc3p-AviTag or Smc3p-BirA and CARs. First, the levels of biotinylated Smc3p-AviTag reporter by the free nuclear BirA at CARs was very similar in S phase and mid-M arrested cells (*Figure 5C, D*), consistent with the robust binding of cohesin to CARs in HU and mid-M arrested cells reported previously (*Blat and Kleckner, 1999*). Second, the CARs continued to bind both Smc3p-AviTag and Smc3p-BirA in mid-M as evidenced by ChIP (*Figure 5—figure supplement 2*). These results suggest that cohesin clusters at arm CARs are downregulated in mid-M by mechanisms other than dissociating cohesin from chromosomes.

To test whether the biotinylation levels of these representative loci were reflective of the genome, we analyzed the mid-M sample by ChIP-seq. All peaks of cohesin clusters correlate with previously identified regions of cohesin binding (*Figure 5—figure supplement 1C*). As predicted from the qPCR results, we observed robust *trans*-biotinylation signal by Smc3p-BirA at the centromeres and pericentric regions and much weaker signal or no signal at arm CARs (*Figure 5E, F*, *Figure 5—figure supplement 1C, D*). In contrast, the biotinylation signal was robust at all CARs in cells with the

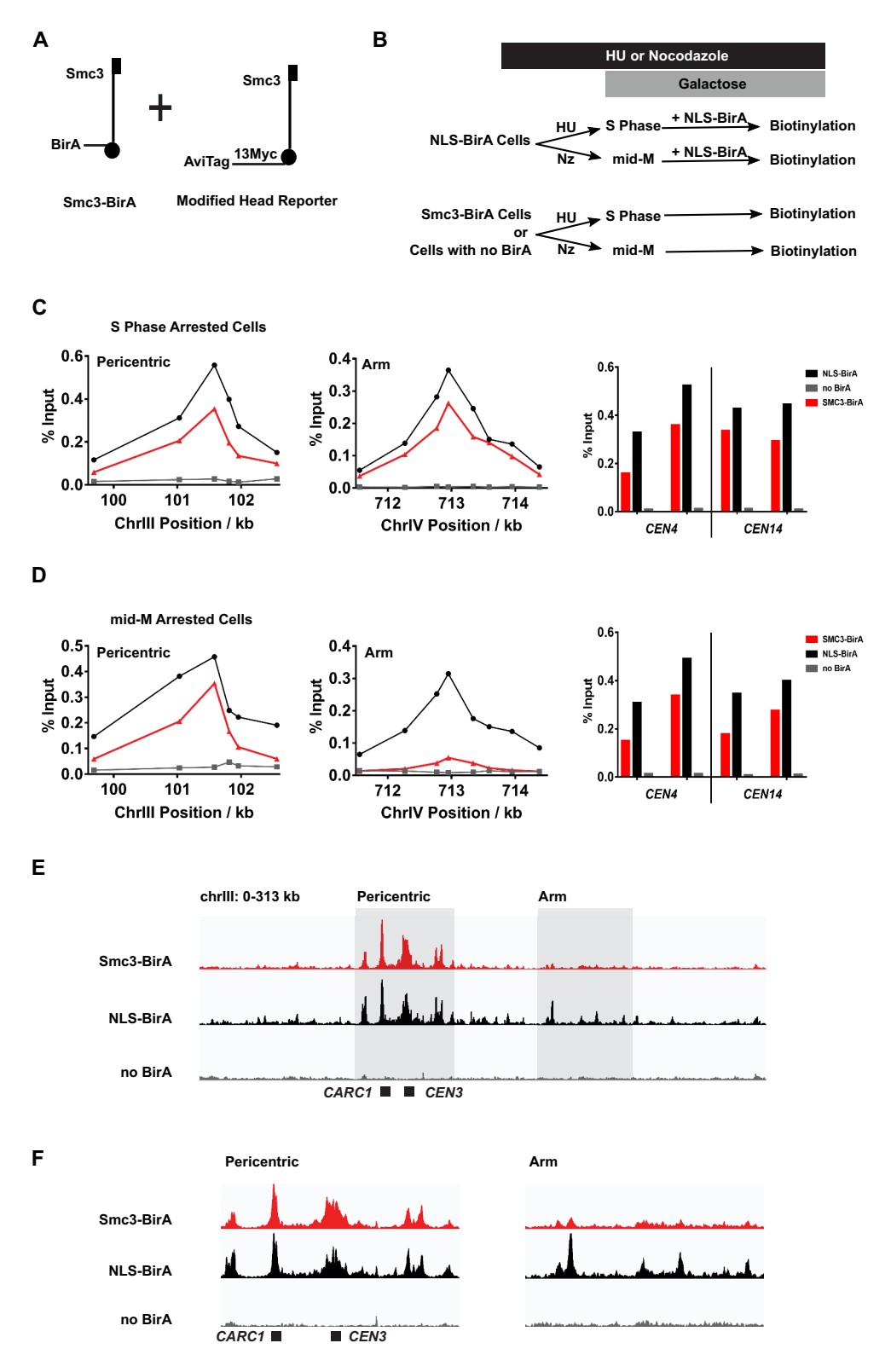

**Figure 5.** Cohesin clusters localize to centromeres and pericentric cohesin-associated regions (CARs) at high levels in both S and mid-M phases, but decrease at arm CARs in mid-M. (**A**) The strain used to assay chromosomal localization of cohesin clusters. The strain carries two *SMC3* alleles (SX155), one expresses Smc3 with BirA in the head domain (Smc3-BirA) while the other expresses a modified Smc3 head reporter with an AviTag fused to Smc3 C-terminus via a 13xMyc linker (Smc3-AviTag). Mcd1p and Scc3p are omitted from the cartoon for clarity. (**B**) Experimental regime used to study

*Figure 5 continued on next page*

*Figure 5 continued*

chromosomal localization of cohesin clusters. Cells were cultured in synthetic media, arrested in S phase (hydroxyurea) or mid-M (nocodazole), and NLS-BirA was induced in the NLS-BirA strain (SX173 expresses NLS-BirA and SX172 has no BirA). Reporter biotinylation was achieved by biotin pulse, and then cells were fixed and harvested for chromatin immunoprecipitation (ChIP) analysis as described in 'Materials and methods'. (**C**) Quantitative analysis of cohesin cluster localization by qPCR in S phase arrested cells. ChIP experiments were carried out as shown in (**B**) with cells arrested in S phase by hydroxyurea. Cohesin clusters in S phase arrested cells were detected on centromeres (right), pericentric CAR (left), and an arm CAR (middle). (**D**) Quantitative analysis of cohesin cluster localization by qPCR in mid-M arrested cells. ChIP experiments were carried out as shown in (**B**) with cells arrested in mid-M by nocodazole. In mid-M arrested cells, cohesin clusters were detected at high levels on centromeres (right) and pericentric CAR (left) but not on arm CAR (middle). (**E**) ChIP to assess cohesin binding to chromosomes. Next-generation sequencing experiments were carried out using samples in (**D**). Biotinylated cohesin clusters were shown in the red trace (SMC3-BirA). Sequencing results of the control strains were plotted at the same scale. The black trace shows the sequencing result from a positive control, where all Smc3 reporter proteins can be biotinylated (NLS-BirA). The gray trace shows sequencing results from a negative control with no BirA. Positions of the centromere and a pericentric CAR were labeled at the bottom of the traces. (**F**) Sequencing traces zoomed in the two highlighted regions in (**E**). The traces on the left show the pericentric region (chrIII: 90–142 kb), and the traces on the right show the chromosome arm (chrIII: 178–237 kb).

The online version of this article includes the following figure supplement(s) for figure 5:

**Figure supplement 1.** Chromatin immunoprecipitation (ChIP) experiments reveal the centromeric and pericentric localization of cohesin clusters.

**Figure supplement 2.** Chromatin immunoprecipitation (ChIP) experiments reveal the centromeric and pericentric localization of cohesin oligomers.

free nuclear BirA. In summary, the qPCR and ChIP-seq results suggest that the majority of cohesins bound to chromosomes are clustering in S phase. In mid-M arrested cells, the presence of clusters remains high at the centromeres and pericentric regions but is reduced on chromosome arms.

## Only fully assembled cohesins cluster

We reasoned that the cell cycle control of cohesin assembly might contribute to cohesin clustering. In G1, only heterodimers of Smc3p and Smc1p form because *MCD1* expression is low and Mcd1p is degraded (*Guacci et al., 1997*). Ectopic expression of Mcd1p in G1 restores Mcd1p levels. Mcd1p binds to the heterodimers and recruits Scc3p, resulting in the formation of fully assembled cohesin (*Haering et al., 2002*). Therefore, we introduced an *MCD1* gene under the control of a galactose inducible promoter (*GAL1)* into our strains with the Smc3p-BirA and the different Smc3p-AviTag reporters, and compared the biotinylation of the reporters in the presence and absence of galactose (*Figure 6A*). In G1 arrested cells, when *MCD1* expression was repressed, a basal level of reporter biotinylation was detected (*Figure 6B*, *Figure 6—figure supplement 1A*), but when *MCD1* was overexpressed by the addition of galactose, 35% of reporters were biotinylated (*Figure 6B*, *Figure 6—figure supplement 1A*). In mid-M arrested cells, where cohesin is already in a fully assembled complex, overexpression of *MCD1* induced by galactose did not affect reporter biotinylation, indicating that there was no change in cohesin clustering levels. Thus the absence of Mcd1p in G1 prevents cohesin cluster formation.

To further test the role of Mcd1p for clustering between G1 and M, we used an auxin degron to deplete Mcd1p (*MCD1-AID*) and assayed Smc3p biotinylation. Cells were arrested in G1, auxin was added to the culture, and then cells were allowed to progress from G1 to mid-M arrest in the presence of auxin. This regimen prevented accumulation of newly synthesized Mcd1p-AID in S and M phases (*Figure 6—figure supplement 1E*). We observed only a basal level (8%) of reporter biotinylation in mid-M phase when Mcd1p was depleted compared to three-fold higher level of reporter biotinylation (25%) in wild-type cells (*Figure 6—figure supplement 1D*). Thus the absence of Mcd1p prevents cohesin clustering in G1 and mid-M phases, suggesting that Mcd1p is essential for cohesin clustering.

We tested whether Scc3p was also required for the *trans*-biotinylation of the Smc3p-AviTag reporter by Smc3p-BirA. We assayed the levels of this biotinylation in cells depleted of Scc3p-AID (*Figure 6—figure supplement 1B*) from G1 to M using the same regimen as we used for Mcd1p-AID. We observed a reduction of biotinylation similar to that in the Mcd1p-AID-depleted cells (*Figure 6D*, *Figure 6—figure supplement 1C*). These results suggest that both non-Smc subunits of cohesin are required for cohesin clustering. Furthermore, clustering can occur in the absence of cohesion since clusters formed in G1 arrested cells.

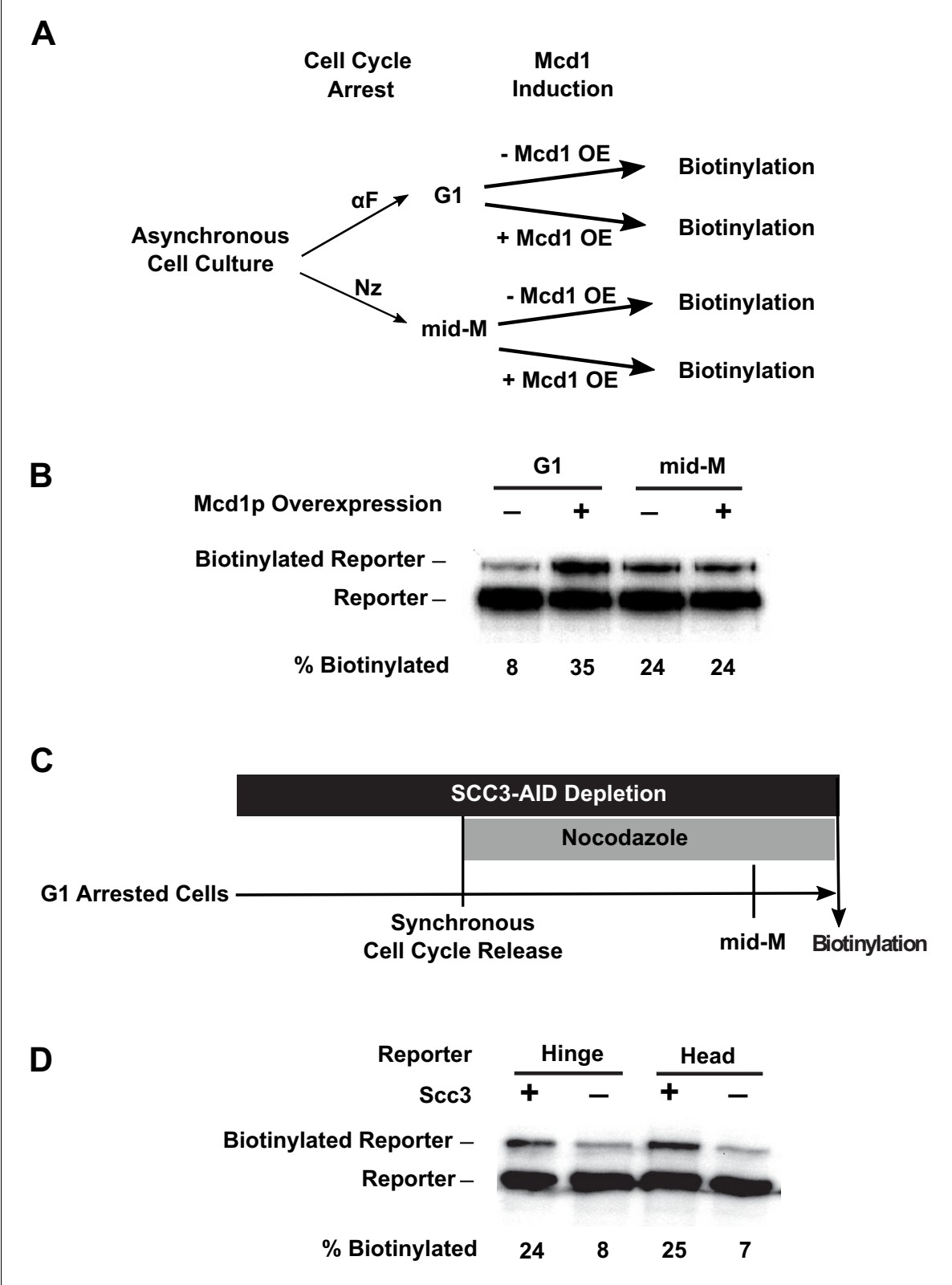

**Figure 6.** Assembly of the full cohesin tetramer is required for clustering. (**A**) Experimental regime used to assess the requirement of Mcd1p in cohesin clustering. The strain carries two *SMC3* alleles with one allele tagged with BirA in the head domain and the other allele tagged with AviTag in hinge domain as shown in **Figure 4A** (SX165). Cultures were arrested in G1 (when Mcd1p is absent) or mid-M (when Mcd1p is assembled in cohesin tetramer). Cultures were split into half, and galactose was added to one aliquot to induce Mcd1p overexpression in arrested cells followed by a short biotin pulse

*Figure 6 continued on next page*

*Figure 6 continued*

to allow proximity biotinylation of the reporters. (B) Effect of Mcd1p overexpression on cohesin clustering. Cells arrested in G1 or mid-M phase were grown with and without Mcd1p overexpression as described in (A). Levels of cohesin clusters were accessed using streptavidin gel shifts as described in *Figure 1D*. (C) Experimental regime used to assess the requirement of Scc3p in cohesin clustering. The strains carry two *SMC3* alleles: one allele was tagged with BirA in the head domain and the other allele tagged with AviTag in the hinge or head domains as shown in *Figure 4A* (SX220B and SX220F). Cells were cultured in low biotin synthetic media, arrested in G1, and depletion of scc3p-AID was carried out by auxin addition. Synchronous cell cycle release was carried out in the presence of auxin, and cells were again arrested in mid-M phase by nocodazole. Proximity biotinylation experiments were carried out in the mid-M phase arrested cells. (D) Scc3p is required for cohesin clustering. Plus signs indicate wild-type cells expressing Scc3p, while minus signs indicate cells depleted of scc3p-AID.

The online version of this article includes the following figure supplement(s) for figure 6:

**Figure supplement 1.** Depletion of SCC2-AID and MCD1-AID.

## Cohesin clustering can occur off chromosomes and in the absence of cohesin acetylation

In budding yeast, cohesin function is modulated by four dedicated regulatory factors, Scc2/4 p, Eco1p, Wpl1p, and Pds5p (*Onn et al., 2008*; *Yatskevich et al., 2019*). We assessed the potential roles of these factors in cohesin clustering by introducing auxin degron alleles or deletion alleles in strains bearing Smc3p-BirA and the Smc3p-AviTag reporters. Cells that were depleted of these factors in G1 by auxin treatment or by deletion were allowed to synchronously progress until arrested in mid-M without the depleted factor. The mid-M arrested cells were assessed for *trans*-biotinylation of Smc3p-AviTag reporter as a readout of clustering.

Scc2p is an essential subunit of the Scc2p/Scc4p cohesin loader and is required for cohesin to bind DNA (*Ciosk et al., 2000*). Eco1p acetylates the Smc3p cohesin subunit, which stabilizes cohesin binding to DNA and enables establishment of sister chromatid cohesion and condensation (*Skibbens et al., 1999*; *Tóth et al., 1999*; *Unal et al., 2008*; *Rolef Ben-Shahar et al., 2008*; *Guacci and Koshland, 2012*; *Çamdere et al., 2015*). We found that depletion of Scc2p-AID or Eco1p-AID had no effect on the *trans*-biotinylation levels seen with either the head or the hinge reporters as compared to wild-type cells (*Figure 7B, C*, *Figure 7—figure supplement 1E, F*). The efficacies of depletion of Scc2p-AID and Eco1p-AID were confirmed by western blot (*Figure 7—figure supplement 1A, C*). Moreover, cohesin failed to bind DNA in Scc2p-AID-depleted cells, confirming its loss of function (*Figure 7—figure supplement 1D*). These results support four conclusions. Neither the loading of cohesin on DNA nor cohesin acetylation is required to form cohesin clusters. Also, clustering occurs in the absence of cohesion as suggested by our experiments analyzing cohesin clusters in G1. Finally, the presence of cohesin clusters alone is not sufficient to ensure cohesion or condensation in budding yeast.

Another regulatory factor called Wpl1p helps dissolve cohesion by removing cohesin from chromosomes, but a subset of cohesin is made refractory to Wpl1p by Eco1p acetylation (*Kueng et al., 2006*; *Chan et al., 2013*). Wpl1p inhibits condensation by an unknown mechanism (*Guacci and Koshland, 2012*; *Lopez-Serra et al., 2013*). In cells harboring a deletion of *WPL1* (*wpl1Δ*), the biotinylation levels of the head or the hinge reporters were indistinguishable from those observed in the wild-type cells (*Figure 7D*, *Figure 7—figure supplement 1G*). These results suggest that Wpl1p is neither an activator nor an inhibitor of clustering.

## Pds5p is required for cohesin clustering on and off chromosomes

The fourth factor, Pds5p, is required for the maintenance of cohesion and condensation in budding yeast (*Hartman et al., 2000*; *Tanaka et al., 2001*; *Wang et al., 2002*; *Noble et al., 2006*). It is also necessary for efficient Eco1p-dependent acetylation of cohesin during S phase (*Chan et al., 2013*). In contrast to other regulators, Pds5p-AID depletion dramatically reduced the *trans*-biotinylation levels of both the head and the hinge reporters (*Figure 7E*, *Figure 7—figure supplement 1H*). The *trans*-biotinylation of the Smc3p reporters without Pds5p was comparable to the basal level observed in G1 arrested cells that lack Mcd1p or mid-M phase cells depleted of Mcd1p (*Figure 4C*, *Figure 6B*, *Figure 6—figure supplement 1D*). Pds5p-AID-depleted cells have reduced cohesin acetylation and DNA binding (*Chan et al., 2013*). However, our analysis of Scc2p-AID and Eco1p-AID depletion cells showed that Smc3p-AviTag reporter biotinylation was independent of either

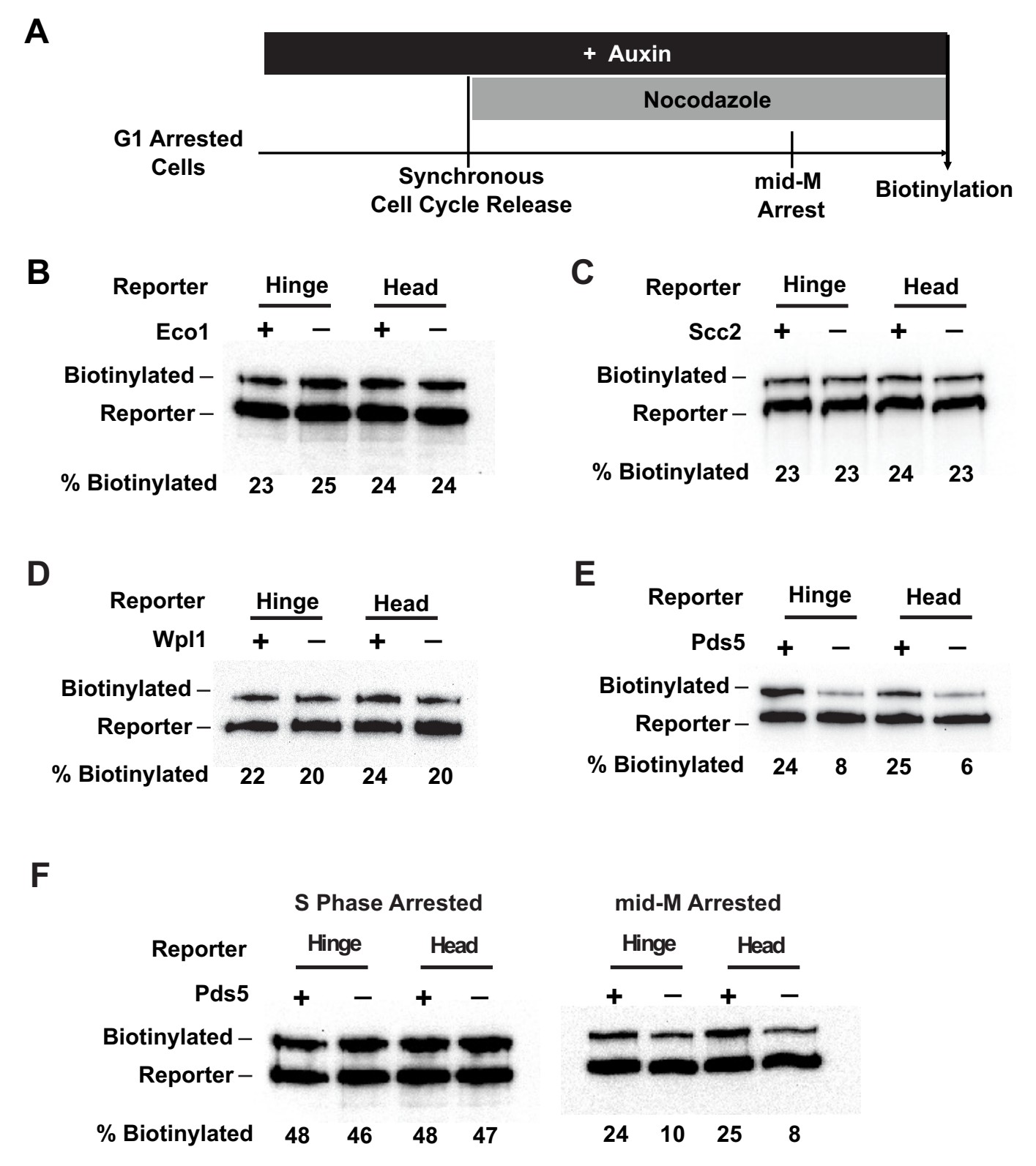

**Figure 7.** Cohesin clustering requires Pds5p, but not Eco1p, Wpl1p, or cohesin loader. (**A**) Experimental regime used to assess the requirement of cohesin regulators for cohesin clusters. Cells were cultured in low biotin synthetic media, arrested in G1, and depletion of the regulator studied was carried out by addition of auxin (except for *wpl1Δ* strains). Synchronous cell cycle release was carried out in the presence of auxin, and cells were again arrested in mid-M by nocodazole. Proximity biotinylation experiments were carried out in the mid-M arrested cells. (**B**) Eco1p is not required for cohesin

*Figure 7 continued on next page*

*Figure 7 continued*

clustering (SX81B and SX81F). Plus signs indicate cells carrying wild-type Eco1p, while minus signs indicate strains depleted of eco1p-AID. eco1p-AID depletion was confirmed by western blotting (*Figure 7—figure supplement 1A*). (C) Scc2p is not required for cohesin clustering (SX82B and SX82F). Plus signs indicate cells carrying wild-type *SCC2*, while minus signs indicate strains depleted of scc2p-AID. scc2p-AID depletion was confirmed by western blotting (*Figure 7—figure supplement 1C*). Cohesin is not loaded onto DNA in the absence of Scc2p (*Figure 7—figure supplement 1D*). (D) Wpl1p is not required for cohesin clustering (SX66B and SX66F). Plus signs indicate strains carrying the wild-type *WPL1* allele, while minus signs represent *wpl1Δ* strains. (E) Cohesin failed to cluster in pds5p-AID depleted cells (SX122B and SX122F). Plus signs indicate cells expressing wild-type *PDS5*, while minus signs indicate strains depleted of pds5p-AID. pds5p-AID depletion was confirmed by western blotting (*Figure 7—figure supplement 1B*). Hinge or head reporters were biotinylated to basal level in the absence of Pds5p. (F) Effects of Pds5p depletion on cohesin clustering in S phase or mid-M arrested cells (SX122B and SX122F). Cells were arrested in S phase or mid-M phase, respectively, and auxin was added to the cell cycle arrested cultures to deplete pds5p-AID. Cohesin clusters were assayed by biotin pulse after pds5p-AID depletion. Cohesin clustering does not require Pds5p in S phase arrested cells but requires Pds5p in mid-M arrested cells.

The online version of this article includes the following figure supplement(s) for figure 7:

**Figure supplement 1.** Depletion of Eco1, Pds5, Scc2, and Mcd1.

acetylation or DNA binding (*Figure 7B, C*, *Figure 7—figure supplement 1E, F*). Therefore, Pds5p likely promotes the ordered cohesin clustering in mid-M by a mechanism distinct from its functions in promoting cohesin acetylation or cohesin binding to DNA.

We wondered whether Pds5p was required for the DNA-independent clustering of cohesins in the absence of Scc2p. To test this prediction, we examined *trans*-biotinylation of the Smc3p reporter in cells depleted of the cohesin loader Scc2p-AID, Pds5p-AID, or co-depleted of Scc2p-AID and Pds5p-AID. When Scc2p and Pds5p were both depleted, the biotinylation levels of both the head or the hinge reporters were reduced to a level comparable to the Pds5p-AID depletion alone and reduced about 70% compared to Scc2p-AID depletion alone (*Figure 7E*, *Figure 7—figure supplement 1J*). These results show that Pds5p promotes clustering of DNA-free as well as DNA-bound cohesin.

## Pds5p is required for the maintenance of cohesin clustering

In these previous experiments, Pds5p could promote the establishment or maintenance of cohesin clusters since Pds5p-AID was inactivated from G1 through mid-M phase. To assess Pds5p's role in maintaining clustering specifically in S and M phases, we depleted Pds5p in cells after they were arrested in S or mid-M. Then we assayed *trans*-biotinylation of the Smc3p-AviTag reporters while cells remained arrested. Pds5p-AID depletion in S phase arrested cells had no effect on cohesin clustering as we found the same high levels of biotinylation on both the head and hinge reporters as seen in wild-type cells (*Figure 7F*, left, *Figure 7—figure supplement 1I*). In contrast, depletion of Pds5p-AID after cells were arrested in mid-M reduced biotinylation of both reporters to basal levels (*Figure 7F*, right, *Figure 7—figure supplement 1I*). These Smc3-AviTag biotinylation levels were similar to the biotinylation levels detected in cells depleted of Pds5p-AID depletion from G1 to M phase. These results suggest that cohesin clusters can be maintained in the absence of Pds5p in S phase, but they require Pds5p to persist post S phase.

The Pds5p-dependent clustering of cohesin in mid-M introduced a possible caveat to our previous interpretation of the *cis*-biotinylation we observed in mid-M in strains with doubly tagged Smc3p containing both AviTag and BirA. Some of the biotinylations might have resulted from biotinylation in trans from BirA attached to a neighboring cohesin in the cluster. To assess the relative *cis*- and *trans*-contributions, we introduced the *PDS5-AID* allele into our set of strains that contain the AviTag and the BirA partners in the same Smc3p. We depleted Pds5p-AID in mid-M arrested cells and then subjected them to a biotin pulse. The levels of biotinylation are slightly lower, but the pattern of biotinylation was the same in the presence of clustering (presence of Pds5p, *Figure 2C*) or absence of clustering (Pds5p depleted, *Figure 7—figure supplement 1H*). These results suggest that the biotinylation patterns we observed with double-tagged Smc3p reflect domain interactions within the same cohesin complex as we previously suggested.

## Evidence that the biological function of cohesin clustering is to maintain cohesion and *RDN* condensation

The requirement of Pds5p in the maintenance of cohesin clusters correlated with its unusual biological functions to maintain cohesion genome-wide and condensation of the tandemly repeated *rDNA* locus. Therefore, we hypothesized that Pds5p-dependent cohesin clustering was necessary for cohesion maintenance and RDN condensation. This hypothesis predicts that cohesin mutations that specifically block cohesion maintenance and *RDN* condensation would likely be defective in cohesin clustering. Only three mutations in yeast cohesin are known to cause defects specifically in cohesion maintenance and *RDN* condensation (*Eng et al., 2014*; *Robison et al., 2018*). Two mutations are in the Mcd1p, one in the linker (*mcd1-V137K*) and the other in the regulation of cohesion and condensation (ROCC) domain (*mcd1-Q266*). The third mutation lies in the Smc3p hinge domain (*smc3-D667*).

To test whether the biological defects caused by these mutations might result from blocking of cohesin clustering, we introduced these mutations into our Smc3p-AviTag reporters and monitored their *trans*-biotinylation by Smc3p-BirA partner. We then assessed for Smc3p reporter biotinylation in mid-M. All three mutations reduced the levels of *trans*-biotinylation of Smc3p-AviTag reporters by 70%, similar to that seen in Pds5p-depleted cells (*Figure 8B, D*, *Figure 8—figure supplement 1*), indicating that they too were defective for the maintenance of cohesin clusters. These results revealed that three domains of cohesin, the Mcd1p linker, Mcd1p ROCC, and Smc3p hinge, are important for cohesin clustering. Furthermore, these results support the hypothesis that cohesin clusters on chromosomes are necessary for maintaining cohesion and RDN condensation.

Differences in the molecular phenotypes of these three mutations provide additional information about the role of cohesin domains in cohesin clustering and cohesion maintenance. Cohesin with mcd1p-V137K binds to chromosomes but fails to bind Pds5p (*Chan et al., 2013*; *Eng et al., 2014*). Failure of the *mcd1-V137K* mutated cohesin in clustering suggests that Pds5p must be recruited to chromosome-bound cohesin to promote cohesin clustering and cohesion maintenance. In contrast, chromosome-bound cohesin molecules with mcd1p-Q266 or smc3p-D667 bind Pds5p (*Eng et al., 2014*; *Robison et al., 2018*), which suggests that recruitment of Pds5p to cohesin is not sufficient for cohesin clustering and cohesion maintenance. Taken together, these results are consistent with a mechanism where Pds5p is first recruited to cohesin by Mcd1p linker, and then the Mcd1p ROCC region, together with Smc3p hinge domain, promotes cohesin clustering and cohesion maintenance. Finally, cohesin molecules with either version of mutated mcd1p subunit are acetylated to the same level as wild-type cohesin and bind to chromosomes as stably as wild-type cohesin (*Eng et al., 2014*; *Chan et al., 2013*). Thus, the mechanism for cohesin clustering and cohesion maintenance is likely independent of any additional functions of these domains in promoting cohesin acetylation and stable binding of cohesin to chromosomes.

## Discussion

Here, we use proximity labeling between cohesin domains to probe the in vivo architecture of cohesin, both on and off DNA. Importantly, labeling only occurs when the two interacting domains are close enough to bring the AviTag to the BirA active site. The proximity labeling of AviTags on the elbow, coiled coil, or the head by BirA on the hinge strongly supports the bending of the coiled coil into a butterfly conformation in vivo, as inferred previously from FRET, genetic experiments, and a recent cryo-EM structure (*Mc Intyre et al., 2007*; *Robison et al., 2018*; *Shi et al., 2020*). However, the existence of only one single butterfly conformation in vivo would not accommodate BirA labeling of multiple AviTags in various positions because of the proximity constraints for labeling by our method. Therefore, we suggest that the labeling of multiple AviTags reflects the formation of additional butterfly conformations. These multiple conformations may reflect the inherent flexibility of cohesin molecules as their coil coils bend to form the structure shown in EM or additional important intermediates in loop extrusion and/or tethering. The formation of multiple distinct butterfly conformations is likely a generic property of SMC complexes as rapid transitions between the ring and multiple butterfly conformations have been observed in recent time-lapse AFM imaging of condensin (*Ryu et al., 2019*).

We also show that BirA partner on DNA could label the head and hinge domains of cohesin, but not the coiled coil. This result is consistent with independent DNA binding activities located in the

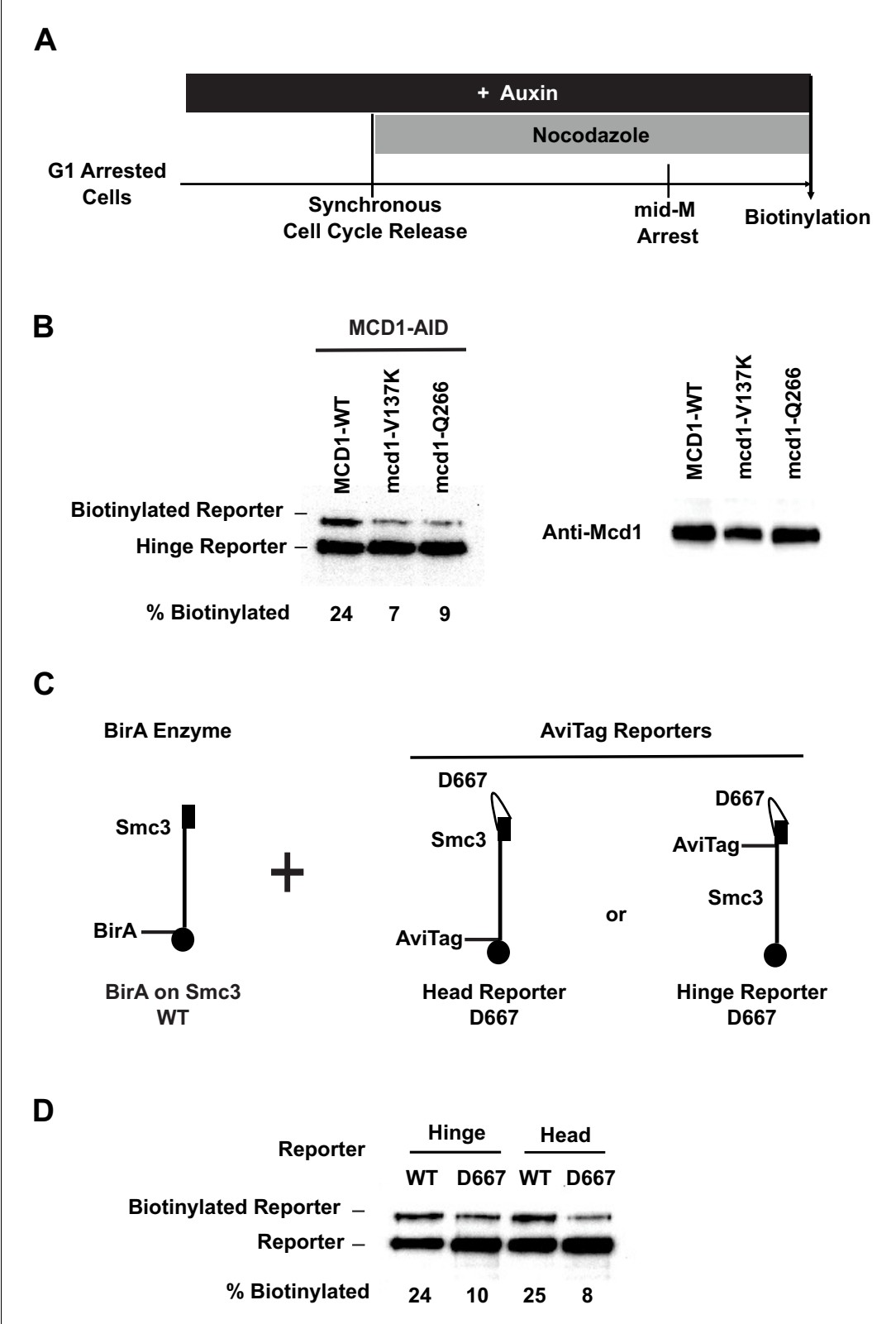

**Figure 8.** Cohesin mutants defective in cohesion maintenance failed to cluster. (**A**) Experimental regime used to assay cohesin clusters in strains expressing wild-type or mutated Mcd1p. Each strain expresses both Smc3p-BirA and Smc3p reporter. Cells were cultured in low biotin synthetic media, arrested in G1, and depletion of wild-type mcd1p-AID was carried out by addition of auxin. Synchronous cell cycle release was carried out in the presence of auxin, and cells were again arrested in mid-M phase by nocodazole with Mcd1p (SX134), mcd1p-V137K (SX130), or mcd1p-q266 (SX129)
*Figure 8 continued on next page*

*Figure 8 continued*

as the sole Mcd1p protein in the cell. Proximity biotinylation experiments were carried out in the mid-M phase arrested cells. (B) Cohesin failed to form clusters in cells carrying either *mcd1-Q266* or *mcd1-V137K* mutation. Each strain carries two *MCD1* alleles: an auxin-depletable *MCD1-AID* and a test *MCD1* mutant allele. Lane 1 has protein extracts from cells expressing wild-type Mcd1p (SX134); lane 2 has protein extracts from cells expressing mcd1p-V137K (SX130); and lane 3 has protein extracts from cells expressing mcd1p-Q266 (SX129). Cohesin clusters were assayed by biotinylation (left), and Mcd1p expression was confirmed by western blotting using Mcd1p antibody (right). (C) Cartoons showing *SMC3* alleles used in (D). Each strain carries two *SMC3* alleles. One allele expresses a wild-type, BirA-tagged version of *SMC3*. The other allele expresses a wild-type or *D667*-mutated version of Smc3p reporter (AviTag inserted in the head in SX49F, and in the hinge domain in SX49B). (D) smc3-D667 reporters failed to cluster with wild-type Smc3p-BirA. Cells depicted in (C) were arrested in mid-M phase, and interactions between wild-type Smc3p and mutated smc3p-D667 reporters were assayed by proximity biotinylation.

The online version of this article includes the following figure supplement(s) for figure 8:

**Figure supplement 1.** Quantification of Smc3 reporter biotinylation in cells expressing *mcd1-V137K*, *mcd1-Q266*, or *smc3-D667*.

head and hinge domains of cohesin. The existence of two independent DNA binding sites in cohesin in vivo has been predicted from mutants that uncouple cohesion from stable DNA binding (*Eng et al., 2014*; *Kim et al., 2019*; *Davidson et al., 2019*). Super-resolution imaging of cohesin on worm meiotic chromosomes has also suggested the proximity of the head and the hinge domains to the chromosomes (*Köhler et al., 2017*). Furthermore, the existence of at least two independent DNA binding activities within cohesin was also inferred from genetic experiments and the observation of cohesin's loop extrusion activities (*Eng et al., 2014*; *Kim et al., 2019*; *Davidson et al., 2019*). The restriction of DNA binding to the head and hinge domains is likely an intrinsic feature of SMC complexes since AFM images of condensin have shown that DNA is localized to the head and hinge of condensin but not the coiled coil.

The existence of butterfly conformations and DNA binding to both the hinge and head domains has been used to propose a scrunching mechanism for loop extrusion by SMC complexes (*Ryu et al., 2019*; *Figure 9A*). These structural features of cohesin could also explain how scrunching could be converted to the DNA tethering that is necessary for cohesion. We previously suggested that cohesin could be converted from extruding DNA to tethering DNA by blocking the release of Adenosine 5'-diphosphate (ADP) from cohesin, thereby trapping an intermediate state in the loop extrusion cycle (*Çamdere et al., 2018*). Here, we speculate that the hinge and head of extended cohesin each could bind a different sister chromatid at or near the DNA replication fork (*Figure 9B*). The subsequent formation of the butterfly conformation, cohesin acetylation, and, possibly, association with other factors prevents ADP release that is needed to dissolve the butterfly. Trapping of this intermediate prevents additional cycles of scrunching that would release the sister DNA molecule, and as a result, generates a stable tether between sister chromatids.

Our biotin-based proximity labeling experiments also revealed that cohesins form ordered clusters in vivo, both off and on DNA. This conclusion is based upon two observations. First, the detection of robust *trans*-biotinylation between cohesin complexes on and off DNA. Second, the pattern of *trans*-biotinylation for different AviTag reporters is more restricted than the patterns of biotinylation by freely diffusible BirA or *cis*-biotinylation within cohesin. Importantly, the restricted *trans*-biotinylation between cohesin, on and off DNA, could not be inferred from a model in which cohesin clustering is mediated by disordered aggregation or DNA-dependent phase separation (*Ryu et al., 2020*). Rather, it fits with the model that cohesin molecules in butterfly conformation form oligomers (dimers or multimers), with the head domain of one cohesin molecule near the head and hinge domains of the other cohesin molecules of the same oligomer.

The existence of ordered clusters of butterfly cohesins on chromosomes in vivo fits several published observations. Cohesin–cohesin interactions on chromosomes in vivo were inferred from functional cooperation between mutated yeast cohesins in vivo (*Kim et al., 2019*; *Zhang et al., 2008*; *Eng et al., 2015*; *Srinivasan et al., 2018*). In addition, a superresolution microscopy study of the worm germline reveals that on the meiotic chromosome axis, meiotic cohesin adopts a conformation that exhibits close proximity between their head and hinge domains (*Köhler et al., 2017*). A second study showed that cohesin clusters (~4) on meiotic chromosomes (*Woglar et al., 2020*). Thus, the clustering of cohesin butterflies may be a conserved feature of eukaryotic chromosomes.

Previously, we suggested two potential roles of cohesin clusters in cohesion (*Eng et al., 2015*). First, cohesion could be achieved by clustering of two cohesins, each bound to a sister chromatid

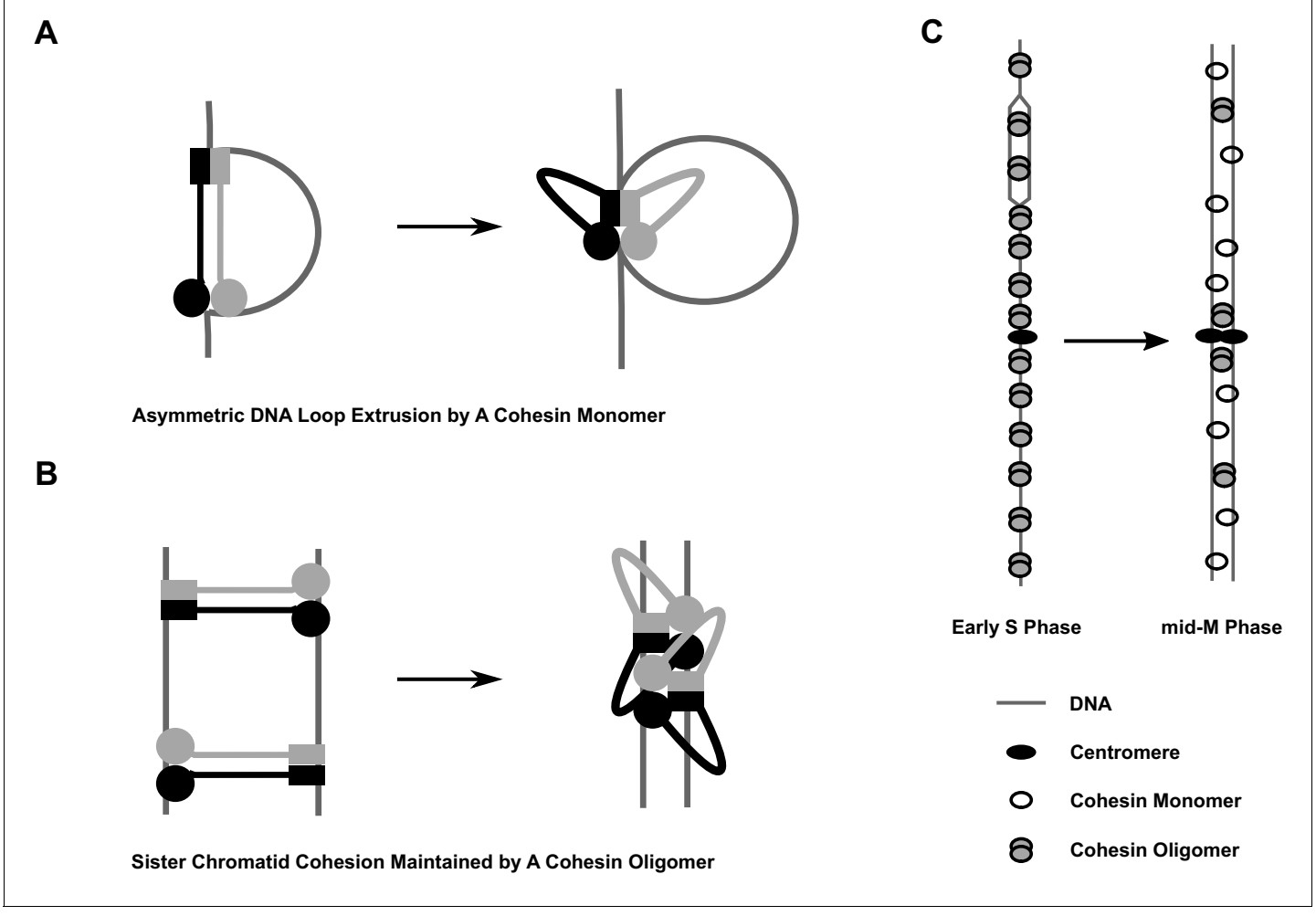

**Figure 9.** Model of cohesin cluster regulation during cell cycle. (**A**) Cartoon showing the initial steps of asymmetric loop extrusion by cohesin. Mcd1p, Scc3p, and Pds5p are omitted from the cartoon for clarity. Cohesin monomer binds DNA with both the head and hinge domains (cohesin complex in extended conformation may adopt rod or open ring structures), then the coiled coil domain bends to form the butterfly conformation and extrude a DNA loop. In subsequent steps (not shown here), DNA is transferred from the hinge to a second DNA binding site on the head. The hinge binds a new region of DNA and repeats the cycle, increasing the size of the loop. Cohesin dimmers can extrude DNA symmetrically. This model has been proposed for the condensin complex (*Ganji et al., 2018*). Mcd1p and Scc3p are omitted from the model for simplicity. (**B**) Cartoon showing sister chromatid cohesion by cohesin. Cohesin binds sisters with its head and hinge domains, respectively, and tethers sister chromatids. This tethering state proceeds to the butterfly conformation where it is trapped. Clustering of cohesins in the butterfly conformation stabilizes tethering between sisters and maintains cohesion. (**C**) Cartoon showing regulation of cohesin clusters during cell cycle. Cohesin complexes form clusters at all previously identified binding sites at early S phase, including centromeres, pericentric cohesin-associated regions (CARs), and arm CAR sites. Most of the clusters at chromosome arms dissolve in mid-M phase. A small fraction of cohesin complexes remain in the clustering state and keep sister chromatids tethered. Pds5 protein participates in the stabilization of cohesin clusters through S phase into mid-M.

(handcuff model, *Zhang et al., 2008*). In this model, clusters would be required for cohesion establishment and maintenance. Alternatively, a single cohesin could have two DNA binding sites, and therefore be competent to form cohesion. Lateral clustering of individually cohesion-competent cohesin would generate sites of multivalent cohesion. In this second model, clustering would not be obligatory for cohesion establishment. Rather, the multivalent cohesion of the cluster would fortify cohesion by making it persist at a specific locus even when tethering of individual cohesins fails. We prefer the second model since in vivo and in vitro studies suggest that individual cohesins have multiple DNA binding sites (this study; *Ryu et al., 2019*), and a single cohesin is competent to generate cohesion (*Haering et al., 2008*).

A role for cohesin clusters in cohesion maintenance is strongly supported by the observation that mutations that specifically fail to maintain cohesin clustering (*pds5-AID*, *mcd1-Q266*, *mcd1-V137K*, and *smc3-D667*) also specifically fail to maintain cohesion (*Hartman et al., 2000*; *Eng et al., 2014*; *Robison et al., 2018*). Furthermore, Pds5p-dependent cohesion maintenance is independent of Eco1p and Wpl1p as is the maintenance of cohesion clusters (*Tanaka et al., 2001*). Pds5p may indirectly promote cluster maintenance by modulating a post-translational modification (other than acetylation by Eco1p) that alters cluster formation or dissolution. Alternatively, Pds5p may act directly to stabilize clusters by binding to different regions of two adjacent cohesin complexes within the cluster. The regions of Mcd1p and Smc3p necessary for cohesin clustering could impact Pds5p function or impact cohesin domains that directly mediate clustering.

The abrogation of cohesin clusters in our mutants also informs on the role of clusters in loop extrusion in budding yeast. We recently showed that cohesin forms positioned loops genome-wide in mid-M. These loops continue to form in cells depleted of Pds5p or expressing mcd1p-Q266 where cohesin clustering is blocked (*Costantino et al., 2020*). The ability of these mutants to loop extrude but not to cluster suggests that cohesin clusters are not obligatory for loop extrusion in vivo. Cohesin clustering, for example, cohesin dimerization, may modulate its ability to promote symmetric or asymmetric looping (*Wang et al., 2017*; *Kim et al., 2019*; *Davidson et al., 2019*).

Our study shows that budding yeast also regulates the level and position of clusters on chromosomes. The ChIP experiment results with *trans*-biotinylated cohesin suggest that clusters occur in S phase at all previously identified sites of cohesin binding (centromeres, pericentromeric regions, and the many CARs along chromosome arms). In mid-M, clusters are specifically reduced at arm CARs (*Figure 9C*). This reduction in arm clusters likely leads to a reduction in the robustness of arm cohesion in mid-M, given the correlation between the cohesin clustering and cohesion maintenance (this study). Interestingly, the reduction of arm cohesion also occurs in mammalian prophase by *WAPL*-dependent destabilization of cohesin binding to DNA (*Kueng et al., 2006*). Thus, the reduction of arm cohesion is likely a conserved feature of mitosis (albeit by different mechanisms).

Proper mitotic chromosome segregation requires the establishment and maintenance of tension that results from the bipolar attachment of sister kinetochores and the surrounding cohesion (*Makrantoni and Marston, 2018*). This cohesion may be strengthened by the cohesin clusters around the centromere, helping the tension to persist after bipolar attachment (*Figure 9C*). Similarly, bipolar attachment of homologs during meiosis I requires tension that is generated by cohesion distal to the crossover site. In *Caenorhabditis elegans*, COH-3/COH-4 cohesins are remodeled prior to anaphase I of meiosis, persisting primarily distal to the crossover (*Severson and Meyer, 2014*). Interestingly, imaging data suggest that COH-3/COH-4 cohesins form clusters. These clusters may also serve to strengthen cohesion and facilitate the persistence of tension. Thus, the regulation of cluster persistence in the genome may be a conserved feature of eukaryotes that is important for proper sister chromatid and homolog alignment in both meiosis and mitosis. In summary, the analysis of cohesin by proximity labeling in vivo has provided important new mechanistic and biological insights into the architecture of the cohesin complex and the cohesin–cohesin interactions.

## Materials and methods

**Key resources table**

| Reagent type (species) or resource | Designation | Source or reference | Identifiers | Additional information |
|---|---|---|---|---|
| Strain, strain background (*Saccharomyces cerevisiae*) | | This paper | | *Supplementary file 1* |
| Sequence-based reagent | qPCR primers | IDT | | *Supplementary file 2* |
| Antibody | Rabbit polyclonal anti-Mcd1 | V. Guacci via Covance | Anti-Mcd1 (555) | WB (1:10,000) ChIP (1:1000) |

*Continued on next page*

*Continued*

| Reagent type (species) or resource | Designation | Source or reference | Identifiers | Additional information |
|---|---|---|---|---|
| Antibody | Rabbit polyclonal anti-Pds5 | V. Guacci via Covance | Anti-Pds5 (556) | WB (1:20,000) |
| Antibody | Mouse monoclonal anti-HA (12CA5) | Roche | 11667203001 | WB (1:8000) |
| Antibody | Mouse monoclonal anti-MYC (9E10) | Roche | 11667203001 | WB (1:8000) |
| Antibody | Mouse monoclonal anti-V5 | Invitrogen | 46-0705 | WB (1:8000) |
| Antibody | Goat polyclonal HRP anti-rabbit | Biorad | 170-6515 | WB (1:8000) |
| Antibody | Goat polyclonal HRP anti-Mouse | Biorad | 170-6516 | WB (1:5000) |
| Other | Protein A Dynabeads | Invitrogen | 10002D | |
| Other | Dynabeads MyOne Streptavidin T1 | Invitrogen | 65601 | |
| Peptide, recombinant protein | Streptavidin | Invitrogen | S888 | |
| Chemical compound, drug | Auxin | Millipore-Sigma | C9911 | 1 mM final concentration, 0.5 M stock in dimethyl sulphoxide (DMSO) |
| Peptide, recombinant protein | Alpha factor | Millipore-Sigma | T6901 | 24 nM final concentration, 10 nM stock |
| Chemical compound, drug | Hydroxyurea | Millipore-Sigma | H8627 | 0.2 M final concentration |
| Chemical compound, drug | Nocodazole | Millipore-Sigma | M1404 | 0.012 mg/ml final concentration, 1.5 mg/ml stock in DMSO |
| Chemical compound, drug | Ethyl acetate ACS | Millipore-Sigma | 319902 | |
| Peptide, recombinant protein | Pronase protease | Millipore-Sigma | 537088 | 10 mg/ml stock in water |
| Other | YNB-biotin | Sunrise Science Products | 1523-100 | |
| Other | BSM powder | Sunrise Science Products | 1387-100 | |
| Chemical compound, drug | D-Biotin | Invitrogen | B20656 | |

## Yeast strains and media

Yeast strains used in this study are MATa cells with A364A background, and their genotypes are listed in *Supplementary file 1*. Synthetic complete with low biotin (SC-Dex) was prepared by dissolving 1.4 g/l YNB-biotin (Sunrise Science Products, see Key resources table), 1.6 g/l BSM powder (Sunrise Science Products), 5 g/l ammonium sulfate (Fisher Scientific), 20 g/l dextrose (Fisher Scientific), and 0.6 nM D-Biotin (Invitrogen) in distilled water and filter sterilized. Synthetic complete medium

with raffinose (SC-Raff) for GAL induction was prepared similarly with SD-Dex, but 20 g/l dextrose was replaced with 20 g/l raffinose (Millipore-Sigma).

## Dilution plating assays

Cells were grown to saturation in YPD media at 30°C, then plated in ten-fold serial dilutions on LEU- or FOA plates as described. Plates were incubated at 30°C for 2 days.

## Synchronous arrest in mid-M phase under auxin depletion conditions
### G1 arrest

Asynchronous cultures of cells were grown in overnight culture to mid-log phase at 30°C in SC-Dex media ($OD_{600}$ = 0.15). Cells were pelleted, then resuspended in fresh SC-Dex media containing 24 nM α-factor (Millipore-Sigma). Cells were incubated at 30°C for 2 hr to induce arrest in G1 phase.

For depletion of AID-tagged proteins, auxin stock was prepared dissolving 44 mg auxin in 0.5 ml DMSO, and 0.05 ml auxin stock was added to each 25 ml medium (~1 mM final). Cells were incubated for an additional 1 hr in α-factor-containing media.

## Synchronous arrest in mid-M phase

About 0.1 mg/ml pronase (Millipore-Sigma, 10 mg/ml stock solution in water) was added to G1 arrested cells, and the culture was incubated at 30°C for 10 min. The cells were spun down and resuspended in SC-Dex with 0.1 mg/ml pronase and 1 mM auxin. Nocodazole (Millipore-Sigma, 1.5 mg/ml in DMSO) was added to the culture dropwise to 12 μg/ml final, and ethyl acetate was added to a final concentration of 1.2%. Cells were incubated at 30°C for 2.5 hr to arrest in mid-M phase.

## Biotinylation of Smc3p-AviTag reporters

Cells were harvested, resuspended in 0.8 ml SC-Dex with auxin, and transferred to a low-binding Eppendorf tube. The biotinylation reaction was initiated by adding 10 nM D-biotin, incubated on a 30°C heat block for 7 min, and terminated by addition of 0.25 ml 80% TCA (Fisher).

## Protein extracts and western blotting total protein extracts

Cell equivalents of 2 $OD_{600}$ were washed twice in cold phosphate buffered saline (PBS) freshly supplemented with 0.5 mM phenylmethylsulfonyl fluoride (PMSF) (Millipore-Sigma) and resuspended in 0.3 ml lysis buffer (15% glycerol, 100 mM TRIS pH 8.0, and 0.2 mM PMSF). The cell lysate was prepared by bead beating on MP FastPrep 5G Homogenizer at top speed for 1 min. The cell lysate was supplemented with 0.1 ml 4× SDS loading dye (16% SDS, 0.2% bromophenol blue, and 20% β-mercaptoethanol) and heated at 95°C for 10 min. The heated lysate was spun at a bench-top centrifuge at top speed for 3 min, and the supernatant was stored in a freezer.

## Streptavidin gel shifts

About 10 μl protein extract was mixed with 6 μl dilution buffer (1× PBS with 20% glycerol) and 2 μl streptavidin solution (Invitrogen, dissolved at 10 mg/ml in PBS), and incubated at room temperature for 10 min. The samples were loaded onto 8% SDS-PAGE gels, subjected to electrophoresis, then transferred to polyvinylidene difluoride (PVDF) membranes and analyzed by western blot using standard laboratory techniques.

## ChIP of biotinylated Smc3p

Cells treated with biotin pulse were fixed with 1% formaldehyde at room temperature for 1 hr, and formaldehyde was quenched by incubating with 1 M glycine for 5 min. The cells were pelleted and washed 3× with FA-STD (50 mM HEPES pH 7.5, 150 mM NaCl, 1 mM EDTA pH 8.0, 1% Triton X-100, 0.1% SDS, 1% sodium deoxycholate, and cOmplete protease inhibitors). The cells were suspended in 1 ml of FA-STD and beaten with glass beads twice on MP FastPrep 5G Homogenizer for 40 s. Chromatin was pelleted at 15,000 rpm for 15 min on a bench-top centrifuge, resuspended in 0.3 ml FA-STD, sheared on a Bioruptor Pico (Diagenode) for 10 min (30 s on/off cycling), and cleared by centrifugation at 15,000 rpm at 4°C for 10 min. Chromatin was then supplemented with an additional 1% SDS and heated at 60°C for 5 min. About 900 μl ice-cold FA-STD was added to the tube to cool down the chromatin. The solution was supplemented with 1 mg/ml of BSA Cohn fraction V

(Millipore-Sigma) and 0.1 mg/ml RNase A (Fisher), then incubated with 60 µl Dynabeads MyOne Streptavidin T1 at 4°C overnight. The beads were washed for 10 min at room temperature for each of the buffers listed below: (1) FA-STD with 1% SDS; (2)FA-STD with 3% SDS; (3) FA-STD with 3% SDS; (4) FA-STD with 1% SDS; (5) FA-STD with 1% SDS and 0.25 M LiCl; (6) FA-STD with 1% SDS and 0.5 M NaCl; and (7) FA-STD with 1% SDS. The beads were then resuspended in 0.3 ml ChIP elution buffer (50 mM TRIS pH 8.0, 10 mM EDTA pH 8.0, 1% SDS, and 0.5 mg/ml proteinase K) and incubated on a 65°C thermomixer overnight. The eluted DNA was purified with MinElute PCR purification kit (Qiagen) and analyzed by qPCR (See *Supplementary file 2* for primer list) or NovaSeq (150 bp paired-end, Illumina). All raw and processed sequencing data were deposited at the GEO under the accession number GSE157155.

### Cohesin ChIP

Cohesin ChIP experiments were performed as described previously (*Eng et al., 2014*; *Robison et al., 2018*) with minor modifications. Cells were fixed with 1% formaldehyde at room temperature for 1 hr, and formaldehyde was quenched by incubating with 1 M glycine for 5 min. The cells were spun down and washed 3× with FA-STD. Chromatin shearing was performed on a Bioruptor Pico (Diagenode) for 10 min (30 s on/30 s off, 10 cycles). Immunoprecipitation was performed using polyclonal rabbit anti-Mcd1p (a gift from V. Guacci) antibodies and protein A Dynabeads (Invitrogen).

## Acknowledgements

We give special thanks to Gavin Schlissel for sharing the BirA plasmid and his guidance on chromatin immunoprecipitation of biotinylated proteins. We thank Dr. Vincent Guacci, Dr. Lorenzo Costantino, Dr. Zhouliang Yu, and Kevin Boardman for inputs and discussions. This work used the Vincent J Coates Genomics Sequencing Laboratory at UC Berkeley. This work was supported by the Helen Hay Whitney Foundation (SX) and the National Institutes of Health (grant 1R35 GM-118189-01 to DEK).

## Additional information

### Funding

| Funder | Grant reference number | Author |
|---|---|---|
| Helen Hay Whitney Foundation | | Siheng Xiang |
| National Institute of General Medical Sciences | 1R35 GM-118189-01 | Douglas Koshland |

The funders had no role in study design, data collection and interpretation, or the decision to submit the work for publication.

### Author contributions

Siheng Xiang, Data curation, Formal analysis, Funding acquisition, Investigation, Methodology, Writing - original draft; Douglas Koshland, Conceptualization, Supervision, Funding acquisition, Writing - review and editing

### Author ORCIDs

Siheng Xiang (iD) https://orcid.org/0000-0003-4475-5707
Douglas Koshland (iD) https://orcid.org/0000-0003-3742-6294

### Decision letter and Author response

Decision letter https://doi.org/10.7554/eLife.62243.sa1
Author response https://doi.org/10.7554/eLife.62243.sa2

## Additional files

### Supplementary files

• Supplementary file 1. Yeast strains used in this study. All strains listed were generated for this study and derived from the A364a background.

• Supplementary file 2. Oligonucleotides used for RT-qPCR.

• Transparent reporting form

### Data availability

Sequencing data have been deposited in GEO as GSE157155.

The following dataset was generated:

| Author(s) | Year | Dataset title | Dataset URL | Database and Identifier |
|---|---|---|---|---|
| Xiang S, Koshland D | 2020 | Chromosome localization of cohesin oligomers in mid-M arrested yeast cells | https://www.ncbi.nlm.nih.gov/geo/query/acc.cgi?acc=GSE157155 | NCBI Gene Expression Omnibus, GSE157155 |

The following previously published datasets were used:

| Author(s) | Year | Dataset title | Dataset URL | Database and Identifier |
|---|---|---|---|---|
| Costantino L, Hsieh TS, Darzacq X, Koshland D | 2020 | Cohesin residency determines chromatin loop patterns | https://www.ncbi.nlm.nih.gov/geo/query/acc.cgi?acc=GSE151416 | NCBI Gene Expression Omnibus, GSE151416 |

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
