## [Decision Letter]

**Acceptance summary:**

This manuscript uses proximity-dependent labelling to explore cohesin dynamics in vivo. The authors observe cohesin clustering that is maximum at S phase and which correlates with cohesion in mitosis. This novel approach allows new insights into cohesin conformations and their functional relevance.

**Decision letter after peer review:**

Thank you for submitting your article "Cohesin in space and time: architecture and oligomerization in vivo" for consideration by *eLife*. Your article has been reviewed by three peer reviewers, including Adèle L Marston as the Reviewing Editor and Reviewer #1, and the evaluation has been overseen by Kevin Struhl as the Senior Editor.

The reviewers have discussed the reviews with one another and the Reviewing Editor has drafted this decision to help you prepare a revised submission.

Summary:

This manuscript uses proximity-dependent biotinylation to probe cohesin structure in vivo. They report 3 main findings. First, the head and the hinge of the same Smc3 protein are in proximity when Mcd1 is present and functional. Second, individual Smc3 molecules which are not part of the same cohesin complex are close enough for biotinylation, even in the absence of stable DNA binding. Third, Pds5 is required for the proximity of separate Smc3 complexes. This manuscript is clearly presented, reports a very promising technique and makes some interesting observations. However, several observations need additional controls and some conclusions need to be toned down to adjust for the limitations of the presented technique.

Essential revisions:

1) Figure 4. The authors find that biotinylated Smc3 is enriched at the pericentromeres and centromeres, but not at arm sites in M phase. One caveat with this experiment is that authors have not shown that the various tagged Smc3 proteins localize normally, only that they are sufficient to support cell viability. If Smc3-BirA itself is preferentially localized at the centromeres, this could explain this result. Since there are Smc3 alleles which affect the ATPase activity of the heads and which accumulate at centromeres, this is a real possibility. The authors should analyse the localization of the different tagged versions of Smc3 by ChIP and compare them.

2) The authors conclude that cohesin adopts a "butterfly" configuration because they observed increased biotinylation when placing Avitag and BirA in the hinge and head domains of Smc3. A few points are relevant to this interpretation. First, the authors need to explain better what they mean by butterfly conformation. Second, the AviTag in the Smc3 coiled coil is located at the Smc3 joint (close to the head domain). Even in the folded state (butterfly?), the coiled coil-AviTag is closer to the head than the hinge AviTag. The schemes presented in Figure 1C and the manuscript text are somewhat misleading. The conclusions on the architecture of cohesin need to be revised accordingly and a more realistic scheme of cohesin (and of elbow-folded cohesin) with the position of the AviTags would be helpful for the reader. Third, taking this into account, the fact that DNA-BirA efficiently labels hinge and heads (but not the coiled coil) is thus likely owed to steric effects (not necessarily reporting on cohesin architecture including elbow-folding). Fourth, the authors need a negative control in Figure 2 to ensure that simply putting both tags in the same molecule doesn't generate biotinylation. If the assay reports on the butterfly conformation as the authors propose, the prediction would be that placing them in the cc and either the hinge or heads would prevent (or at least significantly reduce) biotinylation. This needs to be tested.

3) The experimental procedures do not include sufficient detail to understand how exactly the experiments were performed and there are no figure legends for the main figures. In particular it is difficult to understand how biotinylated Smc3 was detected. Was-an anti-Smc3 antibody used that detects all versions of the protein, or is it only the Avitag that is detected, or some other epitope tag? This is particularly important to know for the "*in trans*" experiments were presumably the different versions of Smc3 are detected. In addition, authors need to show molecular weight marks on the western blots.

4) The term oligomerization is misleading as it implies physical interaction while the biotinylation assay merely measures proximity. The term “clustering” or similar seems more appropriate. This interpretation needs to be revised carefully throughout the manuscript

5) The authors argue that cohesin-cohesin "oligomerization" (meaning proximity) is a function of cohesion maintenance. However, they observe biotinylation when the BirA and Avitag are on different Smc3 proteins even in G1 cells when Mcd1 is overexpression and where there would be no cohesion. Therefore, the conclusion that cohesin-cohesin proximity is a function of cohesion is not valid and needs to be revised.

6) The authors demonstrate that oligomers form off DNA but they also argue, using specific mutants defective in these processes, that the roles of cohesin in cohesion/condensation (which must occur on DNA) are necessary for oligomerization. This calls the functional importance of the authors findings into question and the authors need to clarify this point.

7) The conclusion that Pds5 leads to clustering of cohesin independent of DNA/chromosomes is intriguing. However, given point 6, there is a concern that this observation is a limitation of the assay, rather than reporting on the biology. Ideally this should be substantiated by independent confirmation. Can such clustering off the chromosome be observed by imaging, for example with a split GFP system or similar? Whether this clustering has functional relevance would be another important question, however it is not clear how it can be addressed in the absence of more detailed knowledge about the clustering mechanism. At the very least, the potential short-comings of the assay should be discussed.

8) The authors use bacterial HU protein to localise BirA on DNA, and suggest that this leads to the general presence of BirA along DNA. The potential drawback is that this enzyme binds with higher affinity to certain DNA structures like junctions and nicks, as well as introducing torsional changes into DNA. These peculiarities of HU could affect and hence interfere with the readout. Have the authors tested BirA is localised to an endogenous chromatin protein?

---

## [Author Response]

Essential revisions:1) Figure 4. The authors find that biotinylated Smc3 is enriched at the pericentromeres and centromeres, but not at arm sites in M phase. One caveat with this experiment is that authors have not shown that the various tagged Smc3 proteins localize normally, only that they are sufficient to support cell viability. If Smc3-BirA itself is preferentially localized at the centromeres, this could explain this result. Since there are Smc3 alleles which affect the ATPase activity of the heads and which accumulate at centromeres, this is a real possibility. The authors should analyse the localization of the different tagged versions of Smc3 by ChIP and compare them.

We performed extra ChIP experiments to confirm that BirA-tag and AviTag-tag do not affect genomic localization of cohesin. Cells expressing both AviTag-tagged and BirA-tagged SMC3 were arrested in either early S phase or mid-M phase, fixed and distribution of BirA-tagged and AviTag-Tagged cohesin complexes were analyzed by ChIP-qPCR. The results were added as Figure 5—figure supplement 2. Cohesin binding was assayed at a pericentric binding site (CARC1), a binding site at chromosomal arm (TRM1) and two centromeres. In either S phase or mid-M arrested cells, both AviTag-tagged and BirA-tagged cohesin are robustly detected at these sites at similar levels to wild-type Smc3p.

2) The authors conclude that cohesin adopts a "butterfly" configuration because they observed increased biotinylation when placing Avitag and BirA in the hinge and head domains of Smc3. A few points are relevant to this interpretation. First, the authors need to explain better what they mean by butterfly conformation.

We now define butterfly conformations as any cohesin conformation that involves folding of the coiled coils to bring the hinge and head in closer proximity than the distance in a structure with fully extended coiled coils. The published cryo-EM structure is one of several possible butterfly conformations. We clarify the nomenclature in the Introduction. We believe our evidence now suggests that cohesin structure is likely dynamic accommodating multiple butterfly conformations consistent with the dynamically folding condensin seen in the recent AFM studies.

Second, the AviTag in the Smc3 coiled coil is located at the Smc3 joint (close to the head domain). Even in the folded state (butterfly?), the coiled coil-AviTag is closer to the head than the hinge AviTag. The schemes presented in Figure 1C and the manuscript text are somewhat misleading. The conclusions on the architecture of cohesin need to be revised accordingly and a more realistic scheme of cohesin (and of elbow-folded cohesin) with the position of the AviTags would be helpful for the reader.

A cartoon depicting the elbow folded was added to Figure 2—figure supplement 1A, with positions of AviTag insertion sites labeled.

Third, taking this into account, the fact that DNA-BirA efficiently labels hinge and heads (but not the coiled coil) is thus likely owed to steric effects (not necessarily reporting on cohesin architecture including elbow-folding).

We did not mean to imply that failure to biotinylate the coiled coil reflects a failure to bend at the elbow. We have performed experiments with AviTags in the hinge, elbow, head proximal coiled coil and head, coupled with BirA in either the head or hinge. We have changed the text to clarify the interpretation of the patterns of biotinylation, as shown below:

"The structures of Smc heterodimer and cohesin tetramer are highly dynamic in vivo.

To assess the intramolecular interactions between domains of Smc3p, we constructed five strains. Each strain carried a single SMC3 gene that encoded a doubly tagged Smc3p with one of its domains marked by an AviTag and another by a BirA partner (an example is presented in Figure 2A). […] These results are consistent with the conclusion that the fully assembled cohesin complex, like the heterodimers, is capable of forming multiple butterfly conformations in vivo."

Fourth, the authors need a negative control in Figure 2 to ensure that simply putting both tags in the same molecule doesn't generate biotinylation. If the assay reports on the butterfly conformation as the authors propose, the prediction would be that placing them in the cc and either the hinge or heads would prevent (or at least significantly reduce) biotinylation. This needs to be tested.

See our response to the third question above. The elbow reporter is not biotinylated by the head BirA, showing that not all *cis* reporters are biotinylated efficiently (Figure 2C, fourth lane).

3) The experimental procedures do not include sufficient detail to understand how exactly the experiments were performed and there are no figure legends for the main figures. In particular it is difficult to understand how biotinylated Smc3 was detected. Was-an anti-Smc3 antibody used that detects all versions of the protein, or is it only the Avitag that is detected, or some other epitope tag? This is particularly important to know for the "in trans" experiments were presumably the different versions of Smc3 are detected. In addition, authors need to show molecular weight marks on the western blots.

There was a section in the Materials and methods (“Streptavidin gel shifts”) describing how biotinylation was detected. In the Results section we failed to inform the reader that the sites of insertion contain an HA tag as well as an AviTag. We now describe the assay better in the Results, figure legend of Figure 1C and D and Materials and methods.

"After the biotin pulse, protein extracts were prepared, and the biotinylated and non-biotinylated Smc3p-AviTag was detected by Western blot using the HA epitope. The biotinylated Smc3p-AviTag was present as a slower mobility species due to its binding to the streptavidin presented in the protein sample buffer (Figure 1C and D)."

We did not image the size markers on Western membranes. We do not feel it is necessary to record markers when we are focusing on one protein and the protein (with or without gel shift) generates the only bands on the gel. It is a common practice in the field to leave the marker out (for one of many examples, see Haering et al., 2002). See Figure 1D for the protein markers from our first gel shift experiment.

4) The term oligomerization is misleading as it implies physical interaction while the biotinylation assay merely measures proximity. The term “clustering” or similar seems more appropriate. This interpretation needs to be revised carefully throughout the manuscript

We modified the manuscript to use the term cluster instead of oligomer throughout the text.

"Cohesin may form disordered clusters or ordered clusters (like oligomers) in vivo. If clustering occurs by either mechanism, two domains from different cohesin complexes should be proximal to each other."

However, as we point out that the pattern of differential biotinylation must reflect the formation of ordered clusters and not random clusters.

"The *trans-*biotinylation of the CC reporter by the BirA in the head of another cohesin molecule remained low in S and mid-M phase arrested cells. […] These results are consistent with cohesin clustering in an ordered manner that enables head-head and head-hinge interactions between different cohesin molecules, but disfavors interaction between CC of one cohesin complex and the head of another cohesin."

5) The authors argue that cohesin-cohesin "oligomerization" (meaning proximity) is a function of cohesion maintenance. However, they observe biotinylation when the BirA and Avitag are on different Smc3 proteins even in G1 cells when Mcd1 is overexpression and where there would be no cohesion. Therefore, the conclusion that cohesin-cohesin proximity is a function of cohesion is not valid and needs to be revised.

From comments 5-7, it is clear that the reviewers missed a major point of the manuscript that obviously reflected our poorly written text. This statement is the opposite of what our data suggest and what we intended to convey to the reader. As the reviewer indicated oligomers form independent of cohesion since they form off of DNA, and on unreplicated DNA in S or G1. However, our data strongly suggest that clustering is required to maintain cohesion from S to M. We had suggested this conclusion in the Discussion of the previous version of the manuscript but failed to explain how our results support this conclusion in the Results section. Our logic is the time-tested and central strategy of molecular biology, to look for a correlation between mutations that block a particular molecular phenotype and the appearance of a specific biological phenotype. We show that 4 different mutations that block cohesin clustering block the maintenance of cohesion. We conclude that these results strongly suggest that cohesin clustering is required for cohesion maintenance. We apologize to the reviewers for a poor text that confused them for this central point. We have retitled the last section of the Results and modified the text, to make clear that clustering is required for cohesion maintenance and not cohesion causes clustering.

"Evidence that the biological function of cohesin clustering is to maintain cohesion and RDN condensation.

The requirement of Pds5p in the maintenance of cohesin clusters correlated with its unusual biological functions to maintain cohesion genome-wide and condensation of the tandemly repeated rDNA locus. Therefore, we hypothesized that Pds5p-dependent cohesin clustering was necessary for cohesion maintenance and RDN condensation. […] This hypothesis predicts that cohesin mutations that specifically block cohesion maintenance and RDN condensation would likely be defective in cohesin clustering."

6) The authors demonstrate that oligomers form off DNA but they also argue, using specific mutants defective in these processes, that the roles of cohesin in cohesion/condensation (which must occur on DNA) are necessary for oligomerization. This calls the functional importance of the authors findings into question and the authors need to clarify this point.

See 5.

7) The conclusion that Pds5 leads to clustering of cohesin independent of DNA/chromosomes is intriguing. However, given point 6, there is a concern that this observation is a limitation of the assay, rather than reporting on the biology. Ideally this should be substantiated by independent confirmation. Can such clustering off the chromosome be observed by imaging, for example with a split GFP system or similar? Whether this clustering has functional relevance would be another important question, however it is not clear how it can be addressed in the absence of more detailed knowledge about the clustering mechanism. At the very least, the potential short-comings of the assay should be discussed.

See 5. The fact that cohesin has activities like ATPase or clustering off of DNA does not make them irrelevant for their function on DNA. Indeed our assay may be detecting cohesin dimers. I think the reviewer would agree that forming dimers off of DNA does not exclude the importance of a dimer on DNA as has been postulated for loop extrusion.

8) The authors use bacterial HU protein to localise BirA on DNA, and suggest that this leads to the general presence of BirA along DNA. The potential drawback is that this enzyme binds with higher affinity to certain DNA structures like junctions and nicks, as well as introducing torsional changes into DNA. These peculiarities of HU could affect and hence interfere with the readout. Have the authors tested BirA is localised to an endogenous chromatin protein?

It is not clear to us how HU binding to a specific DNA structure would cause it to be able to label the hinge and head AviTags but much less efficiently the elbow and coiled coil. Also HU binding to DNA does not change local DNA structure (Hammel et al. Sci. Adv. 2016; 2 : e1600650).